


# Correction of wind bias for the lidar on-board Aeolus using telescope temperatures

Fabian Weiler[1], Michael Rennie[2], Thomas Kanitz[3], Lars Isaksen[2], Elena Checa[3], Jos de Kloe[4], Ngozi Okunde[4], Oliver Reitebuch[1]

[1]Deutsches Zentrum für Luft- und Raumfahrt, Institut für Physik der Atmosphäre, Oberpfaffenhofen, Germany
[2] European Centre for Medium-Range Weather Forecasts, Shinfield Park, Reading RG2 9AX, United Kingdom
[3] European Space Agency-ESTEC, Keplerlaan 1, Noordwijk NL-2201 AZ, The Netherlands
[4] Royal Netherlands Meteorological Institute (KNMI), Utrechtseweg 297, 3731 GA De Bilt, The Netherlands

*Correspondence to*: Fabian Weiler (Fabian.Weiler@dlr.de)

**Abstract.**

The European Space Agency satellite Aeolus provides continuous profiles of the horizontal line-of-sight wind component at a global scale. It was successfully launched into space in August 2018 with the goal to improve numerical weather prediction (NWP). Aeolus data has already been successfully assimilated into several NWP models and has already helped to significantly improve the quality of weather forecasts. To achieve this major milestone the identification and correction of several systematic error sources was necessary. One of them is related to small temperatures fluctuations across the 1.5 m diameter primary mirror of the telescope which cause varying wind biases along the orbit of up to 8 m/s. This paper presents a detailed overview of the influence of the telescope temperature variations on the Aeolus wind products and describes the approach to correct for this systematic error source in the operational near-real-time (NRT) processing. It was shown that the telescope temperature variations along the orbit are due to changes of the top-of-atmosphere short- and long-wave radiation of the Earth and the response of the telescope's thermal control system to that. To correct for this effect ECMWF model-equivalent winds are used as bias reference to describe the wind bias in a multiple linear regression model as a function of various temperature sensors located on the primary telescope mirror. This correction scheme has been in operational use at ECMWF since April 2020 and is capable of reducing a large part of the telescope-induced wind bias. In cases where the influence of the temperature variations is particularly strong it was shown that the bias correction can improve the orbital bias variation by up to 53 %. Moreover, it was demonstrated that the approach of using ECMWF model-equivalent winds is justified by the fact that the global bias of models u-component winds w.r.t to radiosondes is smaller than 0.3 m/s. However, this paper also presents the alternative of using Aeolus ground return winds which serve as zero wind reference in the multiple linear regression model. The results show that the approach based on ground return winds only performs 10.8 % worse than the ECMWF model-based approach and thus has good potential for future applications for upcoming reprocessing campaigns or even in the NRT processing of Aeolus wind products.



## 1 Introduction


The European Space Agency (ESA) Earth Explorer satellite Aeolus was successfully launched into space in August 2018 with an intended mission lifetime of three years (Kanitz et al., 2019; Reitebuch et al., 2020). Aeolus, built by Airbus, is equipped with the first-ever functioning space-borne Doppler Wind Lidar (DWL) instrument ALADIN (Atmospheric LAser Doppler INstrument) and provides globally distributed vertically-resolved wind measurements from the ground up to 30 km (ESA,

1999; Reitebuch, 2012a). It measures the component of the wind vector along the instrument's line-of-sight by emitting ultraviolet (UV) laser pulses into the atmosphere and detecting the frequency shifted backscatter signal from molecules and particles (ESA, 2008). The main goal of Aeolus is to improve numerical weather prediction (NWP) by filling gaps for global wind measurements in the Global Observation System of the World Meteorological Organization (WMO), especially in the tropics and the southern hemisphere (Andersson, 2018; Stoffelen et al., 2005, 2020). Another goal is to improve our

understanding of the atmospheric dynamics, especially in the tropics. As spin-off data products, Aeolus provides continuous information about aerosol and cloud distribution, including vertical profiles of backscatter and extinction coefficients (Ansmann et al., 2007; Flamant et al., 2008).

On 9 January 2020, the operational assimilation of Aeolus observations into the European Centre for Medium-Range Weather Forecasts (ECMWF) NWP system was started (Rennie and Isaksen, 2020), followed by other weather centres around the

world, such as the German weather service DWD (Deutscher Wetterdienst), Météo-France and UK Met Office. A prerequisite for this major milestone was the quick identification and correction of the two most important systematic error sources of the Aeolus wind measurements (Reitebuch et al., 2020; Rennie, 2018). The first one was linked to dark current anomalies, so-called "hot pixels", on the Aeolus detectors which cause systematic wind errors of up to several m/s. This issue could successfully be mitigated on 14 June 2019 by applying appropriate correction methods based on dedicated dark signal

calibration measurements (Weiler et al., 2020). The second one was linked to unexpectedly large systematic errors which strongly vary with geolocation. Thanks to the collaborative effort of the teams within the Aeolus Data Innovation and Science Cluster (DISC) (Reitebuch et al., 2019) and the first discovery at ECMWF (Rennie and Isaksen, 2020), a strong correlation of the wind bias with temperature variations across the primary mirror of the telescope could be identified as the root cause of the latter issue.

The small primary mirror temperature variations of 0.3°C, which are related to varying outgoing top of atmosphere (TOA) radiation and corresponding response of the primary mirror's thermal control to that, lead to varying wind errors along the orbit of up to 8 m/s. Thermal changes of the instrument's components along the orbit were already expected before launch. However, it was assumed that these variations would be of smaller error magnitude and would be mainly of harmonic orbit-related nature (Reitebuch et al., 2018b). As the telescope-induced bias turned out to be strongly scene-dependent and not

perfectly harmonic, bias correction tools, which were developed before the Aeolus launch, could not be applied to the



discovered complexity. As a consequence, a new bias correction method using the temperatures of the Aeolus telescope from the Aeolus housekeeping telemetry as independent variables was developed and has been successfully implemented into the operational near-real-time (NRT) processing chain of Aeolus since 20 April 2020.

This article aims to provide a detailed overview of the influence of the telescope temperature variations on the Aeolus winds and the method to correct for the telescope temperature induced wind bias. Section 2 of the paper briefly describes the measurement principles of Aeolus, the design and thermal control of the telescope as well as the Aeolus wind data products. Section 3 depicts the telescope induced wind bias and explains the bias correction method in detail. Section 4 demonstrates the performance of the wind bias correction scheme based on a case study with special settings for the telescope temperatures and also shows the reliability of the method when applied to the complete observation period of six months using Aeolus data

from the first data reprocessing campaign from June 2019 to December 2019. The paper concludes with a summary and an outlook for further analysis.

## 2 Instrument and datasets

This section gives an overview of the measurement principle of ALADIN followed by a description of the instrument's telescope and its thermal control. Finally, the Aeolus data products and variables necessary for this study are presented. For

more detailed information about the instrument, please refer to ESA (2008), Reitebuch et al. (2018a), Reitebuch (2012a) or Lux et al. (2021).

### 2.1 ALADIN configuration and measurement principle

Aeolus flies in a sun-synchronous dusk/dawn orbit at a mean altitude of 320 km with a repeat cycle of 7 days. The satellite carries one single payload, the direct-detection Doppler wind lidar which is pointing toward the Earth under a 35° off-nadir

angle towards the dark side of the terminator (Reitebuch, 2012a). The instrument consists of three main components, the laser transmitter, the telescope and the receiver unit. The ultra-violet laser transmitter emits nanosecond-pulses with a pulse repetition frequency of 50.5 Hz and an energy of ~60 mJ into the atmosphere (Lux et al., 2020a) where the light is scattered on air molecules, aerosol and cloud particles (Reitebuch, 2012b). The backscattered light from the atmosphere is collected by a Cassegrain-type telescope which consists of two mirrors. The primary mirror collects the light and the secondary mirror

reflects it through a hole in the primary mirror to the receiver unit where the Doppler frequency shift of the backscatter light is analyzed. The receiver combines a Fizeau interferometer (FIZ) to analyze the narrow spectral bandwidth backscatter signal from aerosols and cloud particles (Mie channel) and two sequential Fabry-Perot interferometers (FPI) (Rayleigh channel) to measure the broad bandwidth backscatter signal from molecules. The Mie channel uses the fringe-imaging technique which is based on measuring the wind speed dependent horizontal displacement of interference patterns (McKay, 1998). The Rayleigh

channel incorporates the double-edge technique which uses two FPIs as spectral filters that are symmetrically placed around transmitted laser wavelength (Chanin et al., 1989; Flesia and Korb, 1999; Garnier and Chanin, 1992). In the presence of a





Doppler frequency shift the Rayleigh spectrum is shifted towards the spectral peak transmission of one of the two filters. Thus, from the contrast between the transmission of two filters the wind speed along the line-of-sight of the instrument can be determined, and after projection to the horizontal plane the horizonal line-of-sight (HLOS) is obtained.

For both channels Accumulation Charged-Coupled Devices (ACCDs) are used to image the output of the spectrometers (ESA, 2008; Weiler et al., 2020). In the 16x16 pixel illuminated imaging area the return signal is integrated over time based on the vertical range gate settings. The integration time is adjustable and can be changed from 2.1 µs to 16.8 µs which corresponds to vertical sampling of 250 m to 2000 m, considering the 35° off-nadir angle of the instrument. Afterwards, the signals of each range gate are binned together and are continuously shifted downwards to the non-illuminated memory zone of the ACCD

which consists of 25x16 pixels where each row corresponds to one range gate. In the memory zone the return signals of 18 consecutive laser pulses are accumulated to so-called measurements with a duration of 0.4 s (~2.9 km horizontal resolution). Afterwards, the accumulated charges are digitized with 16-bit accuracy and converted into numbers of Least Significant Bits (LSB). In the on-ground processing the signal of typically 30 measurements is accumulated to so-called observations with a duration of 12 s which corresponds to a spatial horizontal resolution of 86.4 km (see Figure 1) for a satellite ground track speed

of 7.34 km/s.

## 2.2 Aeolus Telescope

Aeolus is operated in a mono-axial transceiver configuration which means that the same telescope is used to transmit and receive the light. The Cassegrain telescope consists of a parabolic 1.5 m diameter primary "M1" mirror and a convex, spherical 46 mm diameter secondary "M2" mirror attached to three mounting struts (see Figure 1). The main components of the telescope

are made of Silicon Carbide (SiC) and the wave front error, defined as the deviation of the telescope's wave front from the perfect spherical, was determined after the integration to be below 150 nm rms (root mean square) which is within the specification of 340 nm rms (Korhonen et al., 2008). The distance between the M1 and M2 mirror is 1.32 m. A baffle around the complete telescope structure is used to shield the secondary mirror and the mounting struts from direct sun illumination. On the sun-remote side of the satellite the baffle is shortened to reduce mass and air drag. As Aeolus is facing different thermal

conditions on its orbit which influence the thermal stability of the M1 mirror, an active thermal control system is used. The thermal control of the M1 mirror aims to keep the temperature of the M1 mirror stable at a fixed temperature setpoint of 12° C throughout the orbit, using Thermal Control (TC) thermistors located on the back side of the mirror. For additional temperature monitoring further temperature sensors also located on the back side of the mirror, the so-called Accurate Housekeeping Thermistors (AHT), are available. The AHT sensors are not used in the active thermal control system of the

telescope. Measurements of both sensor types are provided for each observations every 12 seconds in the Aeolus data products. The location of the sensors on the M1 mirror is indicated in Figure 2. The sensors TC-TC-23, TC-29 and TC-32 which are mounted on the bottom and lateral shields of the mirror are not indicated in the figure are not used in the thermal control loop of the telescope.





To control the focus of the telescope the thermal control of the struts and the M2 mirror can be adjusted using dedicated heaters.

This allows to change the distance between the M1 and the M2 mirror which affects the focus of the telescope. So-called Instrument Telescope Refocus (ITR) measurements are carried out on a regular basis to determine the best focus w.r.t. the radiometric performance of the instrument and using the spot width on the Rayleigh channel ACCD as a measure for the telescopes focus. During these measurements, the temperature setpoints of the strut's and M2 control thermistors are varied in the range between 6°C to 16°C in order to derive the optimum setpoint with the best focus.


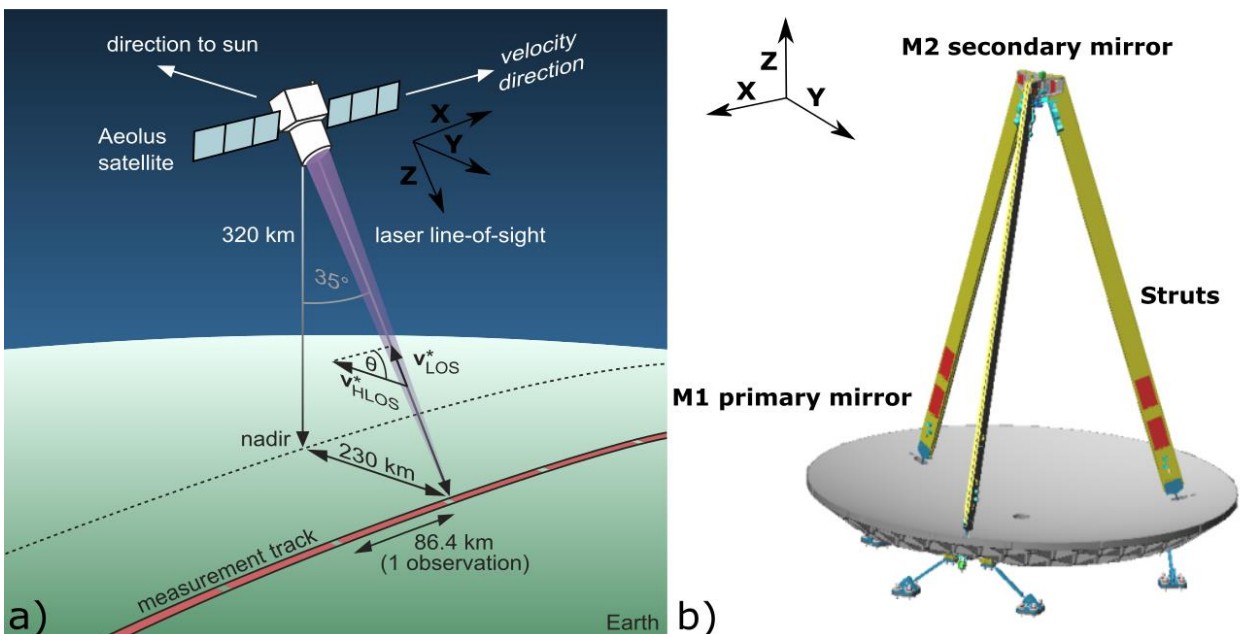

**Figure 1: (a) Aeolus observational geometry (adapted from (Lux et al., 2020b)), and (b) the setup of the Aeolus telescope consisting of the M1 primary and M2 secondary mirrors and the mounting struts (adapted from https://directory.eoportal.org/web/eoportal/satellite-missions/a/aeolus).**




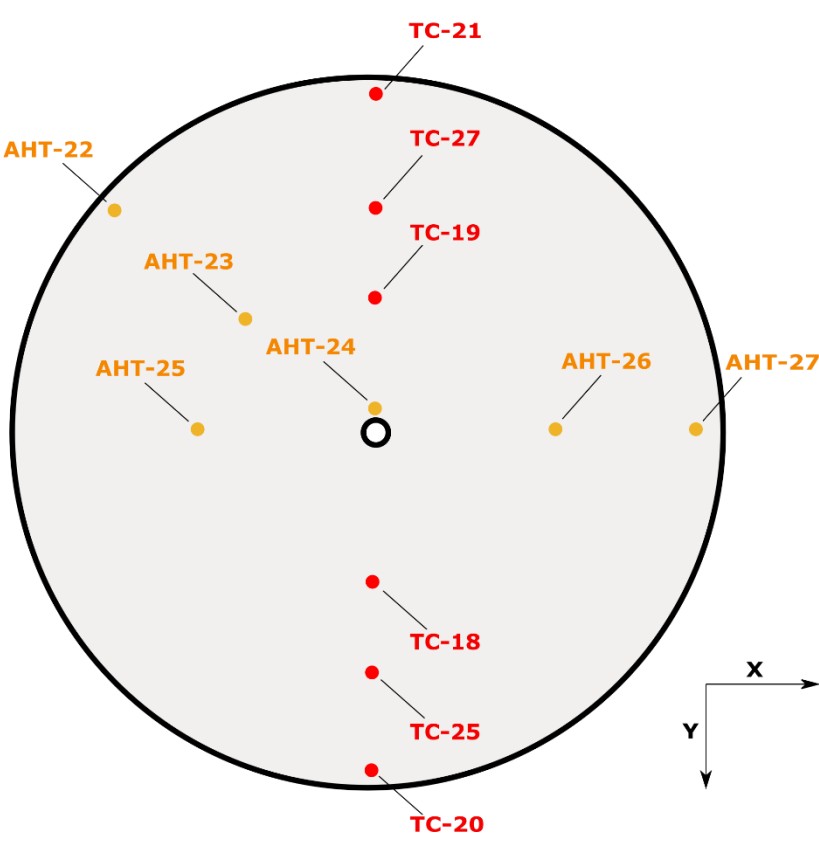

**Figure 2: A schematic illustration of the Aeolus M1 mirror. The red and orange dots indicate the positions of the Thermal Control (TC) and Accurate Housekeeping (AHT) thermistors. X indicates the flight direction.**

**Table 1: Aeolus telescope specifications**

| Parameters | Value |
| --- | --- |
| Type | Cassegrain concept, Silicon Carbide |
| Diameter | Primary mirror M1: 1.5 m, parabolic |
| | Secondary mirror M2: 46 mm, spherical-convex |
| Mass | 67 kg |
| Optical quality | Specification < 340 nm rms Wave-Front error |

## 2.3 Aeolus data products

The Aeolus data processing which is managed by ESA's Payload Data Ground Segment (PDGS) includes several stages to process the raw detector counts up to the main wind product, namely the Level 2B (L2B) data product which contains the

fully-processed horizontal line-of-sight (HLOS) winds for the Mie and Rayleigh channels (Straume, 2018; Rennie et al., 2020).





To continuously improve the quality of the Aeolus data products with a relatively fast cycle for use of Aeolus products in operations at NWP centers, the operational processors are usually updated twice a year. Updates for auxiliary files which are used to control certain settings of the processors, do not follow a fixed schedule and are updated more often. The term "baseline" is part of the PDGS administration and is used to describe a collection of products that have been produced in a
similar way, i.e. using processor versions with similar major version numbers and mostly unchanged algorithm settings.

**Level 1B**

In the L0 and L1A steps, the house-keeping information which consists of various satellite and instrument parameters (e.g. temperatures, pressure, currents, …) is processed and the raw signal data is geo-referenced. The L1B processor provides
processed ground echo data and preliminary winds which are not corrected for atmospheric temperature and pressure influences (Reitebuch et al., 2018a) at the so-called observation level which corresponds to a temporal resolution of 12s corresponding to a spatial horizontal resolution of 86.4 km (see Figure 1). Within the various processing steps, the L1B processor uses a ground detection scheme to flag return signals as ground return signals (Weiler, 2017) with the aim to use these ground detections for zero wind calibration (ZWC). Solid ground is assumed to be a non-moving object and thus can be
used as zero wind speed reference. In addition, this also allows flagging of possible ground affected range bins, since mixing ground and atmospheric backscatter in the same range bin will lead to incorrect wind retrievals. In a first step, the L1B ground detection algorithm identifies ground bin candidates based on a signal-gradient threshold approach (Weiler, 2017). Next, several checks are performed to further restrict the selection of ground bins. For instance, the distance of the ground bin candidates to a model of the Earth's surface is evaluated and also the signal intensity of the ground bin candidates is assessed
to identify valid ground bins. In a final step, the wind retrieval is applied to the valid ground bin signals to retrieve so-called zero wind correction (ZWC) winds (Reitebuch et al., 2018a). The ZWC winds are contained for each channel at observation level in the L1B products and can be used as reference for the M1 bias correction (see Sec. 3.3). The sensitivity of the ground detection algorithm can be controlled by several parameters. For the presented work reprocessed Aeolus L1B data products, processed using the same processor versions as for baseline 1B11 but with custom ground detection settings, from June 2019
to December 2019 were used. For these data products quite relaxed parameter settings for the ground detection were used. The minimum ground useful signal thresholds were set to zero for both channels which leads to a high number of ground returns in the L1B product. This allows for a filtering of the ZWC winds based on the ground return signal before analyzing the relationship between ZWC winds and the M1 temperatures. In the following analysis, a minimum ground useful signal threshold of 9700 LSB is used for the Rayleigh ZWC winds.

**Level 2B**

The L2B processing (Tan et al., 2008) includes a correction of the Rayleigh winds for temperature and pressure broadening effects (Dabas et al., 2008). This correction is based on a-priori temperature and pressure information from short-range





forecasts from the ECMWF weather forecast model. Moreover, the measurement signals (~2.9 km horizontal resolution) from the Mie and Rayleigh channel are classified according to their optical properties based on the scattering ratio and signal-to-

noise ratio (SNR) derived from the Mie channel. This allows the classification of measurements into so-called "clear" (molecular backscatter) and "cloudy" (particulate backscatter) results to avoid contamination from spectrally narrow bandwidth Mie signals in the Rayleigh channel, which would result in errors in the retrieved wind speed (when not accounted for). The classified measurements are then grouped together and horizontally accumulated to optimize the signal-to-noise ratio. The accumulation length is variable and depends on the processor settings and the characteristics of the measurement signals.

Due to the different SNR characteristics of the Mie and Rayleigh signals, the accumulation length is also different for both channels. For the Mie channel, the signals are typically accumulated over a horizontal scale of at most ~10 km whereas the accumulation length for the Rayleigh (clear air) signals is at most ~86 km. It should be noted that for mixed scenes containing both clear and cloudy sections, the accumulation length may be smaller, and this may differ for different altitudes. Afterwards, the wind retrieval is separately applied to each accumulated signal portion to yield two HLOS so-called "wind results" for

each channel: Rayleigh "clear" and "cloudy", Mie "clear" and "cloudy". In general, the focus lies on the Rayleigh "clear" and Mie "cloudy" wind results since they are of better quality. As part of the wind retrieval, a cross-talk correction is applied to the Rayleigh wind results to further minimize Mie contamination (Rennie et al., 2020). Moreover, the L2B products also contain quality flags and wind error estimates for each wind result.

For the presented work Aeolus L2B data products produced by the PDGS, labelled with baseline 2B10, from June 2019 to

December 2019 were used.

**2.4 ECMWF model and O-B statistics**

For monitoring purposes, equivalent HLOS winds from the ECMWF model are calculated for each L2B wind result. For this, information from the Aeolus auxiliary meteorological files (AUX_MET) which amongst many variables contains wind vector information along the predicted Aeolus track is used (Rennie et al., 2020). The information in the AUX_MET file is based on

short range forecasts obtained from the operational ECMWF high-resolution model $T_{CO}1279$ (~9 km grid spacing) (Malardel et al., 2016) usually in the forecast range between 0 h and 12 h. The information in the AUX_MET files is provided every 3 seconds along the orbit at 137 model levels interpolated to the Aeolus track. The nearest-neighbor of the AUX_MET data to the L2B wind results is used to compute observation minus background (O-B) statistics on a global scale. These differences have been used to analyze the systematic and random wind errors of the Aeolus observations at a global scale. The term

background refers to the background model forecast which serves as a priori information for the next analysis run in the data assimilation (Rennie and Isaksen, 2020). From April 20, 2020 (as implemented in L2BP v.3.30, starting with Level 2B products labelled baseline 09) onwards, O-B statistics also have been added to the operational Aeolus L2B products. The first reprocessed dataset from June 2019 to December 2019 also includes this improvement.

For the following analysis of the dependency of the wind bias on the M1 temperatures, a representative average O-B value

"$E(O - B)$" is calculated from the L2B O-B values. As mentioned above, the O-B values are available for each L2B wind



result. However, to decrease the variance of the bias, the O-B values are horizontally averaged to the L1B observation granularity (12 s temporal and 86.4 km horizontal resolution). Afterwards, the O-B values are averaged over all range gates which is justified by the lack of altitude dependency of the M1 bias effect to yield the $E(O - B)$ value. Figure 3 shows the typical distribution of the centre-of-gravity altitudes of the L2B Mie cloudy and Rayleigh clear wind results. This plot indicates

that for the Mie channel large fraction of wind results in the lower altitudes contribute to the $E(O - B)$ statistics. In contrast, the Rayleigh wind results show a broad distribution with equal contribution of the wind results in the altitude range between 5000 m and 18000 m. As explained in the following Sections, the E(O-B) value is used to derive the fit coefficients for the M1 bias correction. Thus, the information from Figure 3 should be kept mind when analyzing the altitude dependency of the error of M1 bias corrected L2B wind results.

It is obvious that the use of the NWP model for the bias correction introduces some NWP model bias dependency, since the ECMWF model wind biases are not zero. However, this is justified by the fact that on average the global bias of the model wind u-components w.r.t reference measurements based on radiosondes is relatively small. This is demonstrated in Figure 4 which shows the time series of daily averages of E(O-B) values of the ECMWF's u-component winds computed for radiosondes and pilots (radiosondes that only measure wind). During all days the bias is clearly below 0.3 m/s, which is

significantly smaller than the Aeolus M1 related bias (for the Rayleigh), justifying the choice of the ECMWF model as reference for the bias correction.

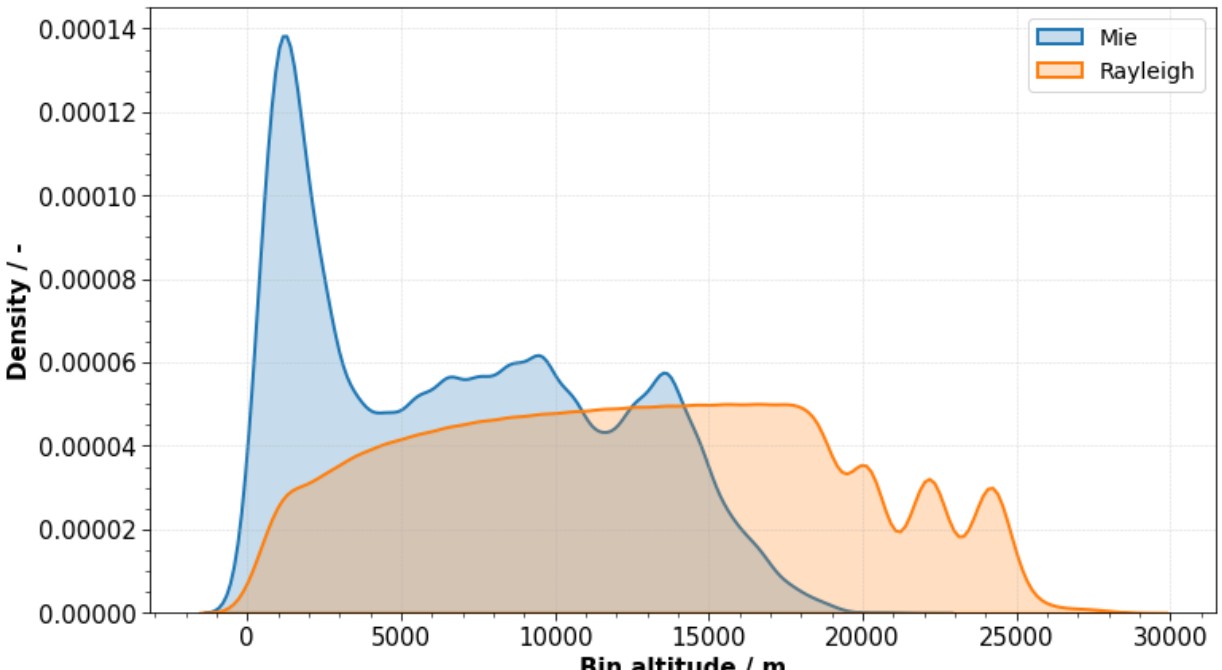

**Figure 3: Density curves of the centre-of-gravity altitudes of the Mie cloudy (blue) and Rayleigh clear (orange) L2B wind results. 74202 Mie and 132042 Rayleigh wind results obtained from 8 August 2019 were used to derive the probability density curves.**


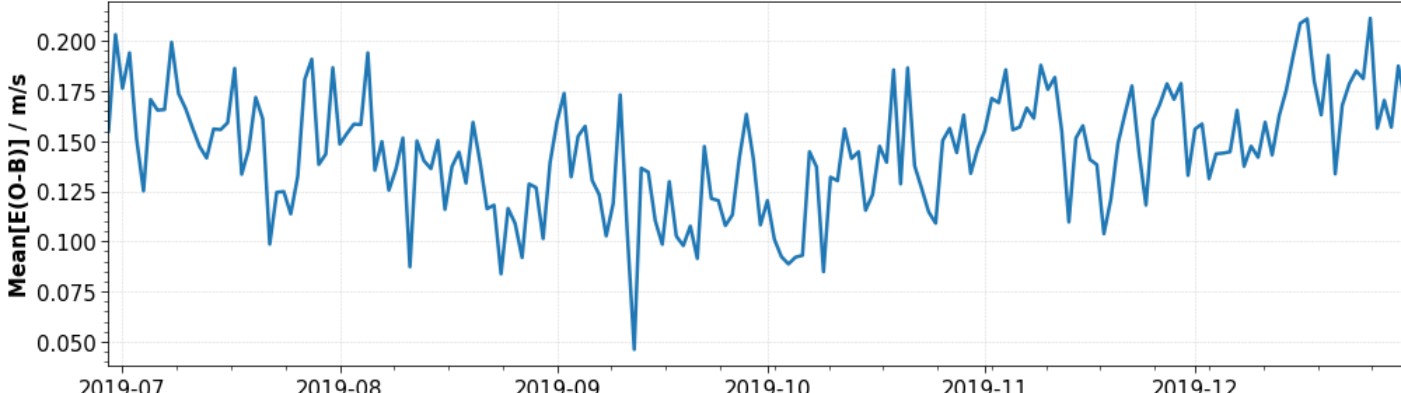

**Figure 4: Time series of global 24-hour averages of E(O-B) values of ECMWF's u-component winds computed for radiosondes and pilots (radiosondes that only measure wind) at all model levels.**

## 3 Methods

The following Section describes the correlation between the wind bias and the M1 telescope temperatures. Moreover, the approach to remove the M1 dependent bias using O-B values and ZWC winds as bias reference is explained and its limitations are discussed.

### 3.1 Telescope induced wind bias

Before launch, harmonic (w.r.t to the orbit phase) bias contributors, induced by thermal effects and pointing variations of the
attitude control system that mainly depend on the latitudinal position of the satellite and the orbit phase, were expected to be dominant. These kinds of error contributors were supposed to be corrected with the so-called Harmonic Bias Estimator (Reitebuch et al., 2018b). This tool was setup before launch based on end-to-end simulations of the assumed harmonic errors using ZWC winds as reference to correct for harmonic bias variations. However, it turned out that this kind of correction was far from being sufficient to correct the bias variation as seen in Aeolus in-orbit data; despite the wind biases having some
harmonic behavior.

Figure 5 shows in-orbit measurements of the Rayleigh clear $E(O - B)$ values as a function of the orbit phase angle (argument of latitude) on 11 August 2019 (blue) and 11 November 2019 (orange). The argument of latitude describes the position of the satellite and is defined as 0° at the ascending node equator crossing and 360 ° at the descending node equator crossing. For both cases the bias shows complex and non-perfectly harmonic dependencies with the orbit phase. Moreover, it was found that
the bias structure changes over the seasons and is strongly dependent on the atmospheric scene, i.e. cloudiness and the TOA temperature. Comparing both cases in Figure 5 shows smaller bias amplitudes in the southern hemisphere in November than





in August. On top of that, the bias shows strong longitudinal dependencies, i.e. non-harmonic elements, which are indicated by the large spread of the bias at a fixed orbit phase, e.g. at around 45° argument of latitude for the August case.

Bell et al. (2008) found strong correlations between housekeeping temperature data and biases for the SSMI/S mission, so easy access to housekeeping data for Aeolus was requested early on during the design of the ground segment. Comparison of the O-B statistics with available housekeeping data then revealed a high correlation of the Rayleigh bias with the M1 temperatures. In particular, a strong linear correlation was found between the Rayleigh bias and the radial temperature gradients of the M1 telescope mirror. This relationship was first discovered at ECMWF and is described in more detail in Rennie and Isaksen (2020). The radial temperature gradient can be described by the following combination of the sensors located at the outer and inner parts of the telescope (see Figure 2): (mean(AHT_27, TC_20, TC_21) – mean(AHT_24, AHT_25, AHT_26, TC_18, TC_19)). Thus, negative values for the radial temperature gradient indicate a warmer central part of the telescope and vice versa.

Further investigations have shown that the orbital variations of the radial temperature gradient are linked to changes of TOA radiation. Figure 6 shows the relationship between the radial temperature gradient measured by Aeolus and the outgoing longwave radiation (OLR) obtained from the NOAA Climate Data Record (CDR) of daily OLR. This dataset is derived from observations from imagers on-board several geostationary satellites such as the High Resolution Infrared Radiation Sounder (HISR) instrument on-board the NOAA19 satellite and provides daily averages of global OLR measurements (Lee and NOAA CDR Program, 2011). OLR measurements from 1 October 2019 were collocated with Aeolus measurements from the same day and averaged along the Aeolus orbit for 60 observations (720 s) resulting in 1388 collocations. It has to be noted that the relationship was found to be not perfectly linear (depicted by the red line in Figure 6) and for other days the non-linearity seems to be stronger. But due to the response of the thermal control of the telescope to the changing environment and the fact that short wave radiation is not considered in the regression analysis, no perfectly linear relation is expected. However, the results depicted in Figure 6 clearly demonstrate the correlation between TOA OLR and changes in the radial temperature gradient of the M1 telescope and motivated further studies of the wind bias correlation with the radial temperature gradient.

To illustrate the relationship between the Rayleigh bias and the mirror temperatures Hovmöller diagrams are generated. Hovmöller diagrams allow to analyze temporal as well as spatial characteristics of a quantity at the same time. Typically, time is plotted on the x-axis and the spatial variable, in this case the latitude of the observation, is used as y-axis. In the Hovmöller diagrams of Figure 7, the mean over all M1 temperature sensors (top), the radial temperature gradients (middle) and the Rayleigh clear $E(O - B)$ values (bottom) are shown at observation level for the complete observation period. It shows that the mean M1 temperature values varies quite remarkably with geolocation and time in the range between 12.9 °C and 14.5 °C. The observed patterns suggest that the variations are due to the changes of the TOA short- and long-wave radiation of the Earth and the response of the thermal control to that. The mean M1 temperatures show a quadrupole-like structure which is visible for ascending as well as descending orbits. Minimum values appear in northern hemisphere summer (June to July) and winter (November to February) in the region of the North and South poles, respectively. Two dominant maximum regions occur in the southern hemisphere between July and November and in the northern hemisphere between September and January.



During polar summer in both hemispheres the high outgoing shortwave radiation fluxes in the polar regions heat up the M1 mirror. This is then compensated by the thermal control system which explains the colder mean M1 temperature values in these regions for these periods. In a similar way the positive anomalies during northern and southern hemisphere winter can be explained. Here, the reduced reflected solar radiation cools down the M1 mirror which is compensated by actively heating

up the M1 mirror. It is remarkable that the intertropical convergence zone (ITCZ) which is characterized by low longwave outgoing radiation is visible in the M1 temperatures. When the satellite passes by the ITCZ, the thermal control reacts by heating the mirror which increases the mean M1 temperatures after the ITCZ. Comparing the mean M1 temperatures between ascending and descending orbits indicates a slight phase shift of the structures between both phases. This can be explained by the inertia of the thermistors and the reaction of the thermal control system. When the satellite crosses the relatively cold ITCZ

it takes some time for the thermal control to respond. As a result, the ITCZ appears slightly shifted to the north and south for ascending and descending orbits, respectively.

The middle plot of Figure 7 shows the radial temperature gradients (as introduced above) along the M1 mirror. Knowing that the central part of the mirror is more exposed to radiation changes, many of the features can be explained such as the difference of the radial gradients between ascending and descending orbits around south-pole during southern hemisphere winter. Here,

the satellite crosses the very cold areas of the southern hemisphere polar vortex which makes the central part of the mirror relatively cold (red colors). Afterwards, on the ascending orbit phase the thermal control responds by heating the central part which results in relatively warmer inner telescope temperatures (blue colors). Comparing the radial temperature gradients (middle) with the Rayleigh clear $E(O-B)$ values (bottom) demonstrates the strong correlation between the bias and the M1 radial temperature gradients. Note that sometimes the bias is due to other issues than changes of the M1 temperatures. For

instance, the striking bias anomalies close the equator for descending paths around mid-August or mid-September are related to the star tracker of the spacecraft being blinded by the moon leading to a wrong determination of the satellite induced LOS velocity and thus, systematic wind errors.

The radial temperature changes of the M1 mirror along the orbit which are up to 0.4°C most likely affect the shape of the mirror. This would change the focus of the telescope and other higher order aberrations and hence, causes slightly different

angular illumination patterns (e.g. incidence angle and divergence) of the light passing through the field stop and illuminating both spectrometers. Both spectrometers are sensitive towards angular changes of the incoming light and thus, produce an apparent frequency shift which manifests as wind bias. Further analysis also shows that radiometric performance of the instrument is affected by the thermal variations of the telescope (Flament et al., 2021).

It should be noted that also sensitivity of the Mie bias on the M1 temperatures was thoroughly investigated. The sensitivity

was found to be ~10 times less than for the Rayleigh channel. This could be explained by the fact that the beam for the Mie spectrometer is increased by a beam expander in front of the Mie spectrometer by a factor of 1.8, and thus reducing the divergence of the light. Furthermore, the Rayleigh spectrometer is specifically sensitive to the incidence angle variation, because of its sequential implementation of the two Fabry-Perot interferometers (Reitebuch, 2012b). Hence, the focus of the analysis below is on correction of the Rayleigh wind bias.


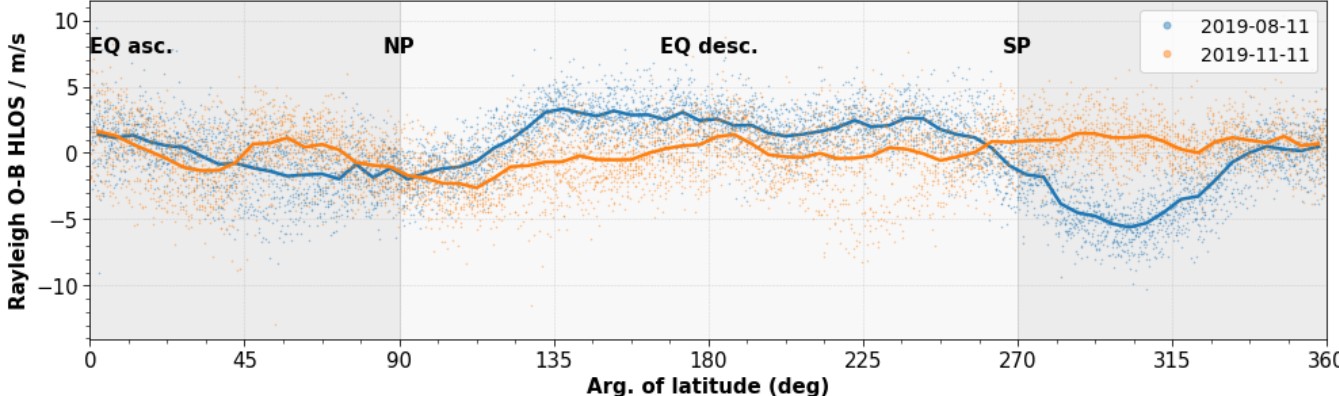

**Figure 5: Rayleigh clear E(O-B) HLOS statistics as a function of the argument of latitude on 11 August 2019 (blue) and 11 November 2019 (orange). The argument of latitude describes the position of the satellite along its orbit in the ascending (asc., dark-grey) and descending (desc., light-grey) orbit phase (EQ: Equator, NP: North Pole, SP: South Pole). The blue and orange dots indicate the E(O-B) values which correspond to averaged O-B values over all altitudes at observation level (introduced in Sec. 2.4). The blue and the orange line show the E(O-B) values as binned average using a bin size of 5° for the argument of latitude.**


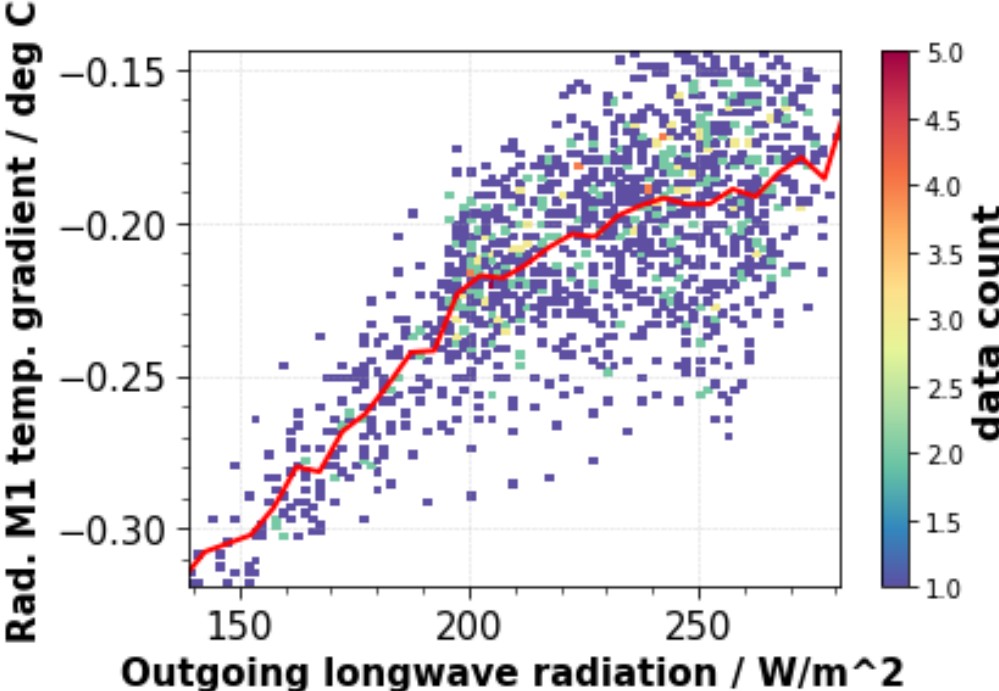

**Figure 6: Correlation between the radial M1 mirror temperature gradient and the daily-averaged TOA-outgoing longwave radiation measured by the HIRS instrument on-board of several NOAA satellites on 1 October 2019 (Lee and NOAA CDR Program, 2011). The red line shows the radial M1 temperature gradient as a function of the outgoing longwave radiation as binned average (5 W/m^2 bin size).**


**Figure 7: Hovmöller diagrams of the average over all M1 temperatures (Top), the radial temperature gradient of the M1 telescope**
**(middle) and the Rayleigh clear E(O-B) HLOS values (bottom) from 28 June to 31 December 2019 split up into ascending (left) and**
**descending (right) orbit phases.**



## 3.2 Bias correction using the ECMWF model

The discovery of the strong linear correlation between the radial gradients of the M1 telescope temperatures (Rennie et al.,
2021) and the wind bias paved the way towards the development of an operational bias correction scheme. For this, a multiple
linear regression (MLR) approach choosing all available thermistors as independent variables is used to describe the E(O-B)
values as a function of the following 15 M1 temperatures: AHT-22, AHT-23, AHT-24, AHT-25, AHT-26, AHT-27, TC-18,
TC-19, TC-20, TC-21, TC-23, TC-25, TC-27, TC-29 and TC-32 (Figure 2). A prerequisite for the operational correction was
the adaption of the L1B and L2B processors to include these variables in the operational data products (since baseline 2B09)
which was a huge achievement by the DISC team given the short amount of time for preparation. With this information the
MLR model can be defined as follows:

$$E(O - B) = \beta_0 + \beta_1 \cdot AHT22 + \beta_2 \cdot AHT23 + \cdots + \beta_{15} \cdot TC32 + \varepsilon \tag{1}$$

where $\beta_0$ is the intercept, $\beta_1 \ldots \beta_{15}$ are the coefficients for each temperature variable and $\varepsilon$ denotes the error term. In terms of
reducing the bias, it turned out that the MLR showed better mathematical performance than a linear model that is based on the
radial physical temperature gradient of the telescope (see Sec. 3.1, (Rennie and Isaksen, 2020)).

Figure 8 demonstrates that the MLR model described in Eq. (1) is a suitable choice for this task. For the diagnosis, in the same
way as done for the NRT-processing, past data, in this case from 11 August 2019, is used to predict the bias on 12 August
2019. Note that for the reprocessing data from the same time period is used to derive the fit coefficients. This even further
improves the performance of the bias correction scheme. The left scatter plot indicates the high correspondence between the
model prediction and the measured bias values. This is demonstrated by the high R² value of 0.78. The right plot of Figure 8
is generally used to indicate if the model residuals show remaining patterns that are not fully captured by the model (James et
al., 2013). This is done by plotting the model residuals against the predicted values. In our case, the residuals are equally
scattered around zero without dominant patterns justifying the MLR approach. There seems to a slight hint for remaining
residual bias (< 0.3 m/s) in the range between 0 m/s to 2 m/s. This is shown by the smooth curve fitted using weighted least
squares (red line) fit to the residuals which shows slightly negative values in this region.

Figure 9 demonstrates the application of the M1 bias correction for 12 August 2019. For this example, the model fit coefficients
$\beta_0, \ldots, \beta_{15}$ are derived from Rayleigh clear E(O-B) values from 11 August 2019 and are used to predict the Rayleigh clear
$E(O - B)$ values on the next day. The plots compare corrected (orange) with uncorrected (blue) $E(O - B)$ values as a function
of time (top) and the argument of latitude (bottom). To measure the performance of the bias correction approach to decrease
the bias variation along the orbit, the standard deviation of the $E(O - B)$ values, i.e. $STD[E(O - B)]$, can be used. In the case
depicted in Figure 9, this value is reduced by 52.8 % from to 2.89 m/s to 1.36 m/s (also see the text box in Figure 8) which
clearly demonstrates how well this method works to reduce most of the M1 induced bias variation along the orbit. The
remaining residual variation of 1.36 m/s is considered to be of random nature arising from instrumental and forecast model





random errors and does not contain any obvious regular patterns. It should be noted that this value of the standard deviation is representative for altitude averages of the E(O-B) value at L1B observation granularity, in contrast to the wind random error which is defined as the standard deviation of each single observation within a vertical profile.

For the M1 bias correction in the operational processing chain, dedicated software was developed which has been put into operations on 20 April 2019 (starting with Level 2B products labelled baseline 09). Figure 10 depicts the flow chart of the M1

bias correction for the Aeolus NRT operational processing chain. The AUX_TEL software uses 24 h of past L2B data as input, performs the MLR (see Eq.1) and writes the model coefficients $\beta_0...\beta_{15}$ into an auxiliary telescope (AUX_TEL_12) file. The generation of the AUX_TEL_12 file is updated every 12 hours. The AUX_TEL_12 file is used as input by the L2B processor for the M1 bias correction of the wind results. This is done by solving Eq.1 using measured M1 temperatures and the derived model fit coefficients to yield a M1 bias correction value for each L2B wind result of both channels. As next step, the bias

correction values are subtracted from the measured wind results and provided in the L2B products as bias corrected winds. Note that bias correction values are also written into the Earth Explorer format L2B product which allows users to undo the M1 bias correction.

The low update frequency of 12 h for the AUX_TEL_12 generation is necessary because the model parameters $\beta_0...\beta_{15}$ are slowly changing with time which indicates that the sensitivity of the instrument towards telescope temperature variations is

changing over time. Moreover, this allows to capture the slowly drifting global average bias. Investigations have shown that the global average bias changes are due to a slow drift of illumination of the Rayleigh spectrometers in the internal path (particularly for the FM-B laser). In order to capture this effect, it was decided to implement an intercept term $\beta_0$ (see Eq. 1) into the model which makes sure that mean of the model residuals, i.e. the $Mean[E(O - B)]$ value of the analyzed period, is zero. To avoid large constant bias offsets from the model, the mean winds are anchored to the ECMWF model twice per day.

To make sure that the sample size is large enough and the fit coefficients are derived with a sufficiently high enough accuracy, 24 hours of past data (~6500 samples) is used in the MLR.

As mentioned before, the Mie-cloudy winds are much less affected by thermal variations of the M1 mirror. However, it was decided to also use the same correction approach for the Mie winds, mainly for the reason to correct for global bias offsets of the Mie-cloudy winds, again related to internal path drifts.






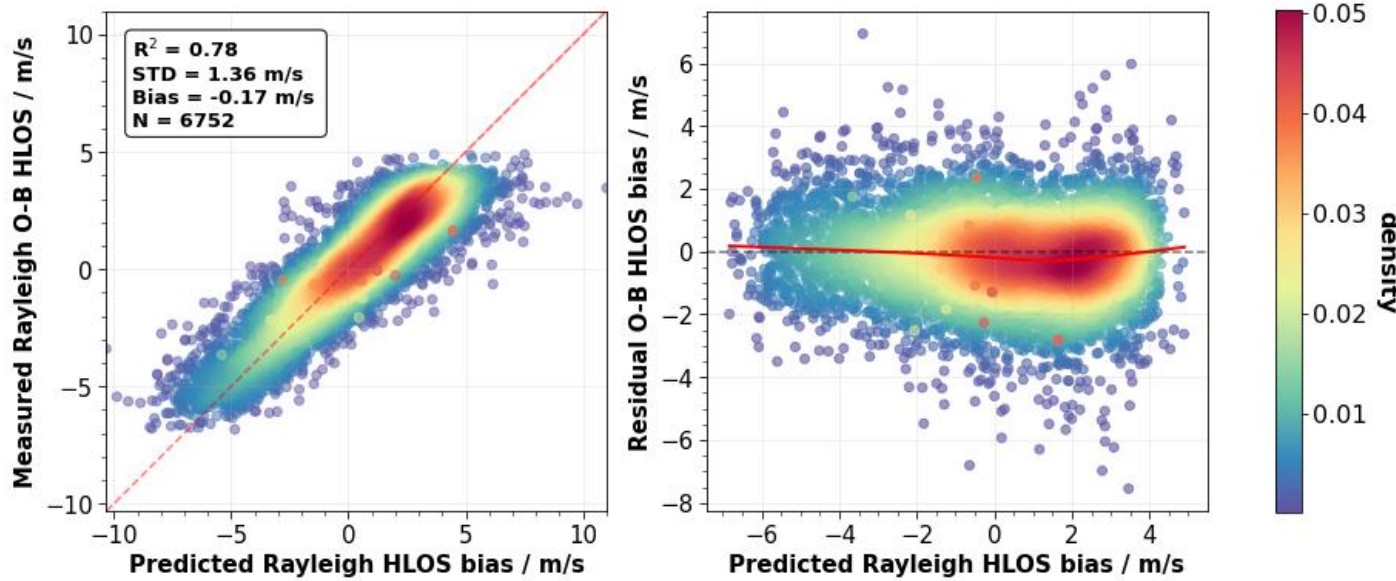

**Figure 8: Diagnostics plots for the multiple linear regression with the measured Rayleigh clear E(O-B) values (left) and the model residual (right) as a function of the predicted bias (left). Data from 11 August 2019 is used to predict the bias on 12 August 2019. (left). The red line in the left plot indicates the diagonal. In the textbox of the left plot values for the coefficient of determination ($R^2$),**
**standard deviation (STD) and the bias of the corrected values as well as the number of data points (N) used in the regression is shown. The red line in the right plot indicates the smooth function obtained after applying locally weighted smoothing. The color-coding in both plots indicates the kernel density.**

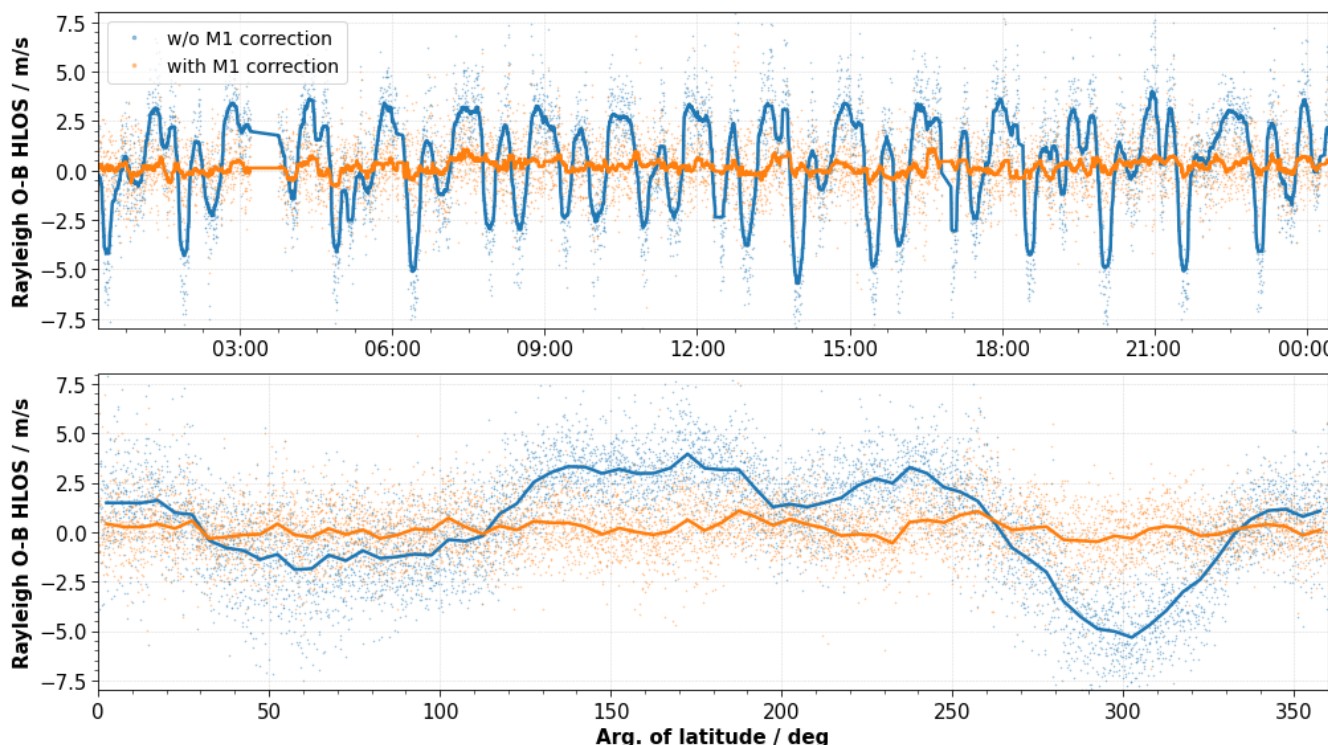

**Figure 9: Rayleigh clear E(O-B) HLOS values as a function of time (top) and the argument of latitude (bottom) during 12 August 2019. The blue and the orange dots indicates the bias without and with M1 bias correction, respectively. The blue and the orange lines in the top and bottom plots show temporal (5 min interval) and binned averages (5° bin size for the argument of latitude) of the E(O-B) values, respectively. The M1 bias correction coefficients are derived from data from 11 August 2019.**





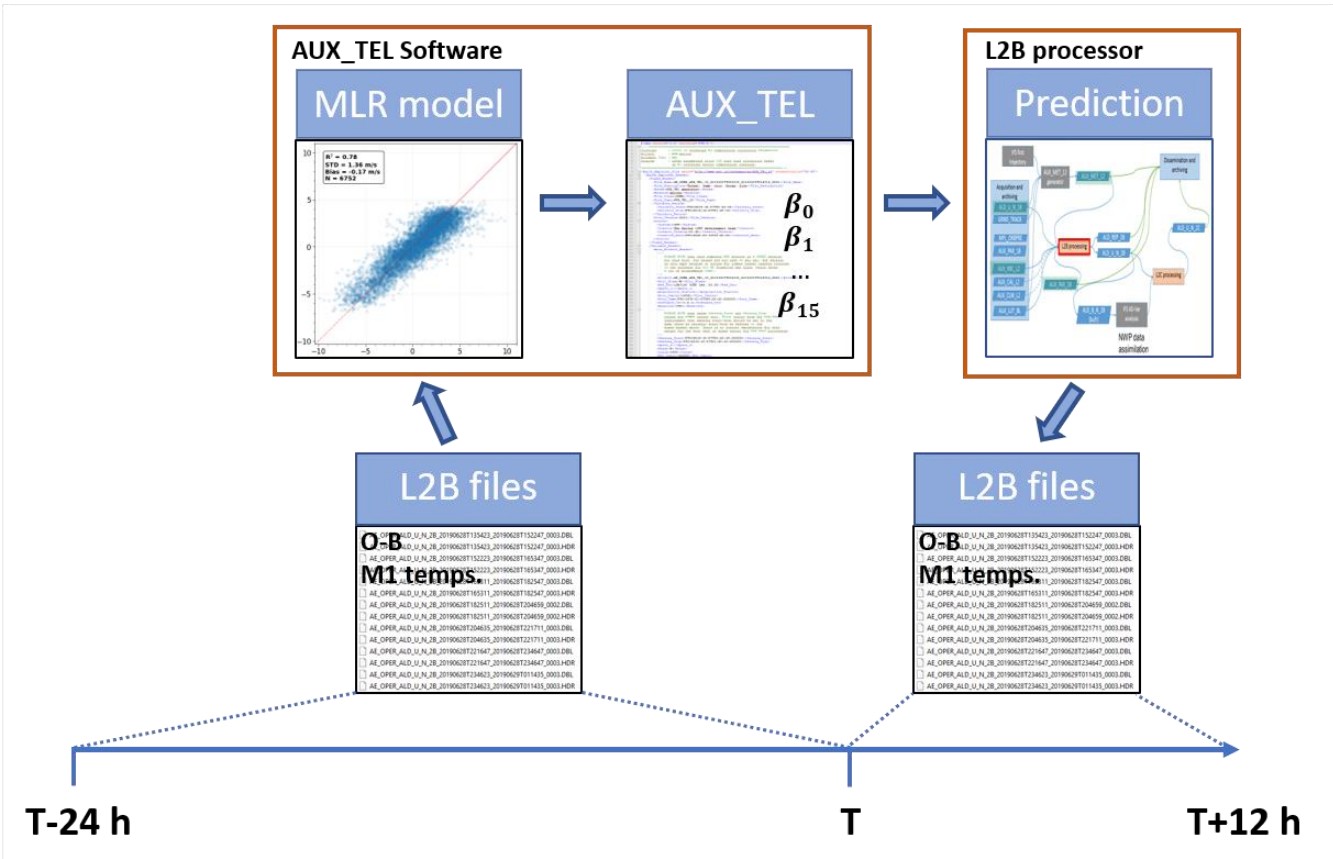

**Figure 10: Flow chart of the operational M1 bias correction of the L2B wind results. In the AUX_TEL software, 24 hours of past L2B data with the E(O-B)-values as dependent and the 15 M1 temperatures as independent variables are used as input for the multiple linear regression (MLR) model. The model software produces an AUX_TEL file which contains the model coefficients $\beta_0 \ldots \beta_{15}$. Afterwards, the L2B processor uses the AUX_TEL files to make a prediction for the wind bias and to correct the wind results of the subsequent 12h window. Then, the AUX_TEL file is updated in the same way.**

### 3.3 Bias correction using ground return winds

The operational M1 bias correction procedure makes use of ECMWF model winds and thus, introduces some dependency on the NWP model. But as mentioned in Sec. 2.4 of the manuscript, this is justified by the low model wind bias w.r.t to radiosondes (see Figure 4). In addition, the M1 bias correction uses 24 hours of global model winds averaged over all altitudes, which is why small-scale model biases (e.g. in the tropics) appear only as an additional noise source in the fit procedure. Moreover, potential altitude-dependent ECMWF model bias is not an issue, as vertically-averaged E(O-B) statistics are used in the MLR model.

However, to overcome this issue of model dependency, it is also possible to use Aeolus' L1B ground return winds (see Sec. 2.3) as reference instead of E(O-B) values. Ground return winds can be seen as zero wind reference to correct for systematic





430   wind error sources such as the M1 temperature induced wind bias. However, the use of ground return winds as reference is
hampered by the limited spatial and temporal coverage of ground returns. The availability of ground returns with high enough
ground signals is mainly restricted to polar regions with high surface albedo. The top plot of Figure 11 which displays Rayleigh
clear O-B HLOS and ZWC winds before the application of the M1 correction as a function of the argument of latitude during
11 August 2019, shows the large difference of the data availability between O-B and ZWC values. In this case, the availability

of winds is mainly restricted to the ice-covered regions around Antarctica which results in a rather small sample size of 659
ZWC winds compared to 6897 O-B values. However, it turned out that the coverage of ZWC winds is sufficiently high, i.e.
enough different O-B values are covered, to use ZWC winds as bias reference in the M1 bias correction. The plot shows a
large correspondence between O-B and ZWC values as both indicate the same M1 dependent bias structure. Note that the
constant offset of about 3 m/s between O-B and ZWC values is due to the different calibration procedure between L1B and

L2B winds and is not considered to be a problem for the bias correction since this offset could be corrected in the data
processing.

In contrast to the MLR model defined in Eq.1 a slightly different approach is used to describe the ZWC winds as a function of
the M1 temperatures. Due to the lower sample size a simplified model with fewer independent variables has to be used. It was
found that a grouping of the thermistors into two groups which describe the temperature at the outer and inner parts of the M1

mirror      provides      the      best      results:      $G1 = mean(AHT27, TC20, TC21)$      and      $G2 = mean(AHT24, AHT25, AHT26, TC18, TC19)$. The bias correction model is then described as follows:

$$ZWC = \alpha_0 + \alpha_1 \cdot G1 + \alpha_2 \cdot G2 + \varepsilon \tag{2}$$

where $\alpha_0$ is the intercept, $\alpha_1, \alpha_2$ are the coefficients for each temperature group G1 and G2 and $\varepsilon$ denotes the error term. For

the M1 bias correction of L2B winds Eq.2 is solved using measured M1 temperatures and the derived model coefficients
$\alpha_1$ and $\alpha_2$ to yield a M1 bias correction value for each L2B wind result. The bottom plot of Figure 11 shows the application of
the ZWC based M1 correction for 12 August 2019. The green curve indicates the Rayleigh clear O-B HLOS values after the
ZWC based M1 correction. To compare both approaches, the O-B bias after the ZWC based M1 correction (green) is shown
together with the operational O-B based bias correction (red). This demonstrates that also the ZWC based approach is capable

of reducing most of the M1 temperature induced bias variation. The ZWC approach reduces the $STD(E(O - B))$ from 2.89
m/s to 1.40 m/s which is only slightly worse than the O-B based approach achieving a reduction to 1.36 m/s. The offset between
both curves is a result of the different calibration procedure between L1B and L2B winds as discussed above. The similar
performance of the ZWC approach also helps confirm that the O-B approach is doing the correct thing and not introducing too
many ECMWF model bias related artefacts.


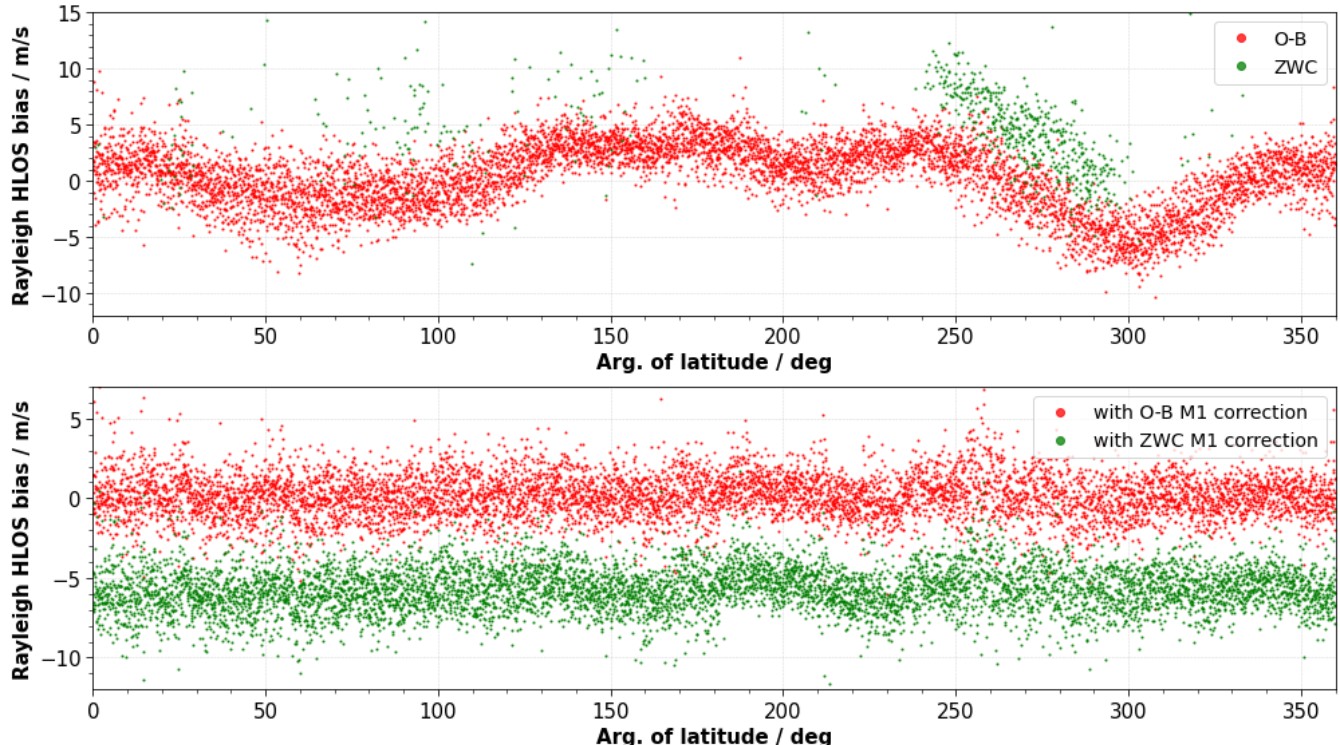

**Figure 11: (Top) Rayleigh clear O-B HLOS values (red) and Rayleigh ZWC HLOS winds (green) and without M1 correction as a function of the argument of latitude during 12 August 2019. (Bottom) The red and green indicate the Rayleigh clear O-B HLOS values after the M1 correction using O-B values and ZWC values as bias reference, respectively. The M1 bias correction coefficients**
**for both approaches (ZWC, O-B) are derived from data from 11 August 2019.**

## 4 Results

In this Section, it is demonstrated that the M1 bias correction also works with different temperature set point conditions for the thermal control thermistors of the primary telescope mirror. Moreover, the performance of the M1 bias correction using
O-B values is evaluated for the completed observation period from 28 June to 31 December 2019, demonstrating the reliability of this method. In addition, the performance of the bias correction using ground return winds is shown.

### 4.1 Case study

In order to decrease the orbital variation of the wind bias and increase the atmospheric return signal, in-orbit tests were carried out to optimize the thermal control of the telescope from 6 to 10 July 2020. The main goal of the tests was to decrease the
orbital variations of the M1 mirror radial thermal gradients by modifying the control law coefficients of the heater lines. Figure 12 (top) shows the evolution of the radial M1 temperatures (see Sec. 3.1) before and during the M1 optimization tests. The radial gradients are shifted towards higher values and also the range increased from -0.3°C – -0.1°C before the test to -0.25°C





– 0.1°C during the test. As a result, also the Rayleigh clear bias (blue line in bottom plot of Figure 12) changed significantly. The plot indicates a decrease of the variability of the bias on sub-orbital timescales but an increase of the bias variability on

longer time scales. However, the orange line that indicates the M1 corrected bias clearly demonstrates the capability of the M1 bias correction approach to also perform well with new temperature settings. In this case, data from the same day is used to derive the fit coefficients. During the optimization test the M1 bias correction improved the standard deviation of the O-B values by 53.2 % from 5.09 m/s to 2.70 m/s. This example also shows how the M1 bias correction removes the global offset introduced by changes of the illumination in the internal path (see Sec. 3.2). In this case, the offset improves on average from

-7.49 m/s to 0.0 m/s.

This important finding implies that the M1 telescope induced bias can be handled by ground processing under any circumstance and further M1 optimization tests can now be fully focused on optimizing the radiometric performance of the instrument.

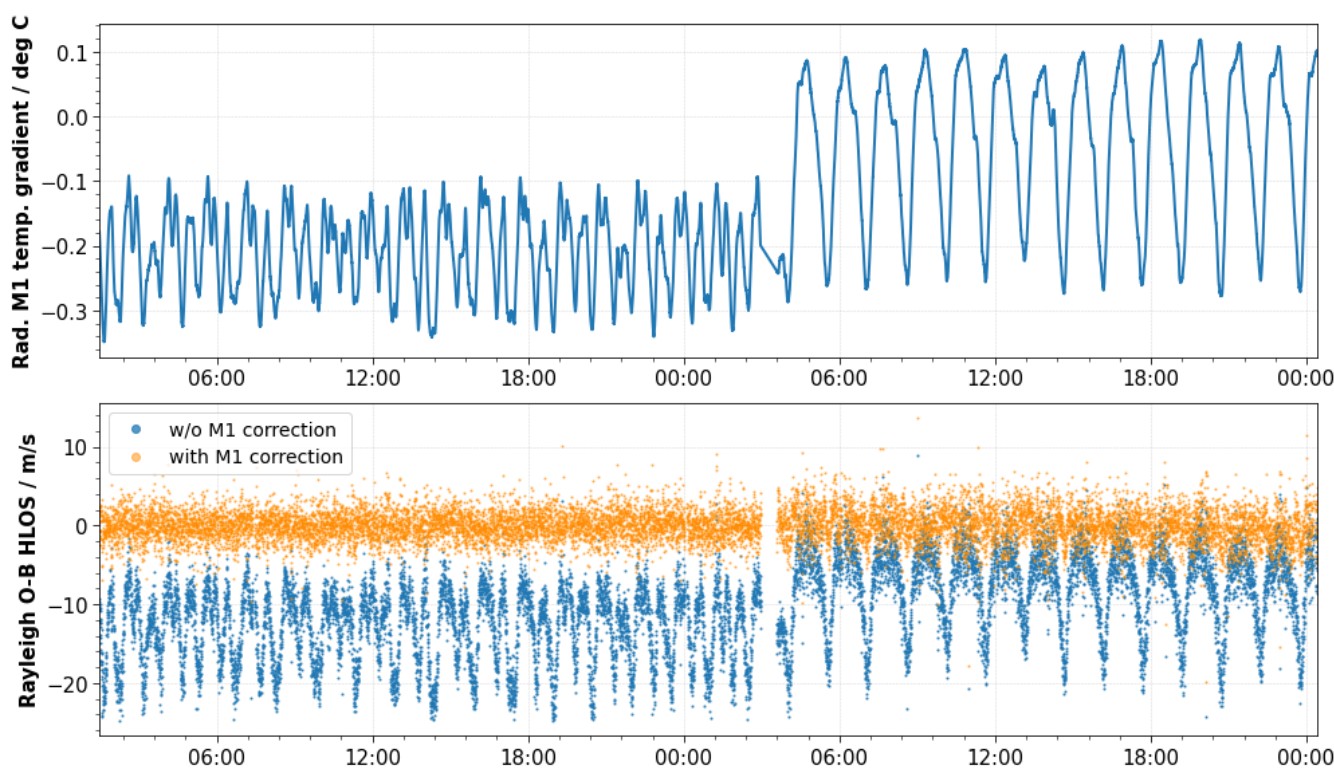

**Figure 12: The radial temperature gradient of the M1 telescope (top) and the Rayleigh clear E(O-B) HLOS values (bottom) as a function of time during the M1 optimization tests on 5 and 6 July 2020. The blue and the orange dots indicates the bias without and with M1 bias correction, respectively. Data from day N is used to predict the bias on day N. The periodicity (especially visible for the second half of the plot) is related to the orbital phase of the satellite which is ~ 91 minutes.**



## 4.2 Performance time series

Figure 13 shows the performance of the M1 bias correction for the period from 28 June to 31 December 2019. To generate the
time series data from day $N$ was used to predict the wind bias on day $N + 1$. The top plot shows the standard deviation of the
Rayleigh $STD(E(O - B))$ values before and the after the application of the M1 correction based on O-B and ZWC values. In
general, this plot shows a changing seasonal influence of the M1 temperature induced Rayleigh bias. At the beginning of the
period from July to October the largest variability of the radial gradients of the M1 temperatures along the orbit can be observed

(see Figure 7) which leads to large M1 temperature induced wind bias and hence, large improvement by both bias correction
approaches. For instance, on 15 July 2019 M1 bias correction drastically reduces the $STD(E(O - B))$ values from 2.29 m/s to
1.24 m/s and 1.37 m/s for the O-B and ZWC approaches, respectively. For the following period between August and October
a steady decrease of the M1 influence on the Rayleigh wind bias can be seen. This is mostly related to a seasonal effect that
decreases the orbital variability of the radial temperature gradients on the descending orbit phase (see Figure 7). In October

and November, the M1 temperatures induced bias variability reaches its minimum. During this period, the influence of the
bias correction on the Rayleigh wind bias is very small. The plateau of increased standard deviation values between 28 October
and 15 November is during a campaign period with special range gate setting with more smaller range gates to achieve finer
altitude resolution around the tropopause, resulting in higher random wind errors. Afterwards, the seasonal effect on the M1
temperature variations slowly starts to increase again.

The difference between the performance of the O-B (red) and ZWC (green) approach is not constant. On average, the ZWC
approach is 10.8 % worse than the O-B-based correction with a maximum deviation of up to 25.6 %. However, one should
bear in mind that the verification against O-B itself will naturally favor the O-B method. Especially, at the end of period, when
the M1 influence on the wind bias is low, the performance of the ZWC approach decreases and is not able to further improve
the $STD(E(O - B))$ values. As a consequence, it was decided to use the ECMWF model as reference for the operational

correction of the NRT products. However, methods to further improve the performance of the ZWC based approach are still
under investigation, which might allow to use this approach for future reprocessing or even NRT-processing of the Aeolus
data products, removing the need for ECMWF model winds as a reference. Other NWP centers have confirmed the low biases
with respect to their own independent NWP models following the operational implementation of the M1 temperature bias
correction, which is reassuring.

It is worth noting that the M1 bias corrected $STD(E(O - B))$ values show a steady increase of 11.3 % from 1.24 m/s at the
beginning to 1.40 m/s (O-B based) at the end of the period. This is due to a combination of decreasing laser emit energy from
65 mJ to 61 mJ and a loss of the optical signal throughput in the receive path of the instrument. Without the implementation
of the accurate M1 bias correction, it would not have been possible to observe the increase of the random error based on wind
error statistics.

The bottom plot of Figure 13 shows the temporal evolution of daily averages of the Rayleigh clear $Mean(E(O - B))$ values
with (red) and without (blue) O-B based M1 correction. As mentioned in Sec. 3.2, the M1 bias correction also removes the





constant offset from the winds which is due to changes in the illumination of the spectrometers for the internal path. The blue curve shows that the daily averages of the $Mean(E(O - B))$ values slowly decreased at different drift rates from +2.1 m/s to -7.5 m/s. To correct for this effect, the M1 bias correction is updated once per day. In the operational processing, the updates

are even performed twice per day which allows for a more accurate and reactive correction of the constant bias offset. After the M1 correction the bias using O-B values (red line) is in the range between +0.9 m/s and -0.6 m/s proving the capability of bias correction to remove the constant offset. Smaller peaks of the corrected bias, e.g. on 15 July or 1 September, are related to larger steps of the bias development and could be avoided by further increasing the update frequency of the bias correction. For the reprocessing, this issue is solved by using data from the same day to derive the MLR coefficients.

Figure 14 shows the global distribution of the Rayleigh clear ($E(O - B)$) values obtained from one week of data from 15 to 22 August 2019 before (top) and after the M1 bias correction (bottom) using O-B values as bias reference. As discussed in Section 3.1, the Rayleigh bias is a complex function of the changes of the M1 radial temperatures gradients and the response of the thermal control. During this week, the temperature variations were particularly strong which explains the strong orbital bias variations in the range between -6 m/s and 8 m/s. However, the M1 bias correction successfully removes latitudinal and

longitudinal bias patterns from the winds and reduced the $STD(E(O - B))$ value for this period from 2.84 m/s to 1.35 m/s. It is important to mention that the M1 bias correction aims at globally removing the average offset, i.e. the $Mean(E(O - B))$, w.r.t to the ECMWF model. The operational correction makes sure that the vertically averaged mean wind bias is removed, i.e. $Mean(E(O - B))$ equals zero. As a consequence, when smaller data samples, e.g. in the framework of localized comparisons between Aeolus and ground-based or airborne measurements, are analyzed, it might be that the bias of M1

corrected Aeolus winds is not zero (Martin et al., 2020; Belova et al., 2021; Guo et al., 2021). This also becomes clear when looking closely at the bottom panel of Figure 14 where the bias of the corrected O-B values, despite showing some residual effects, is in the range between -3 m/s and 3 m/s, depending on the geolocation. However, for the purpose of improving NWP prediction at a global scale this approach is considered to be the best method available at the moment, allowing operational assimilation of the Aeolus wind product and demonstrating positive impact on numerical weather forecast (Rennie and Isaksen,

2020; Rennie et al., 2021).



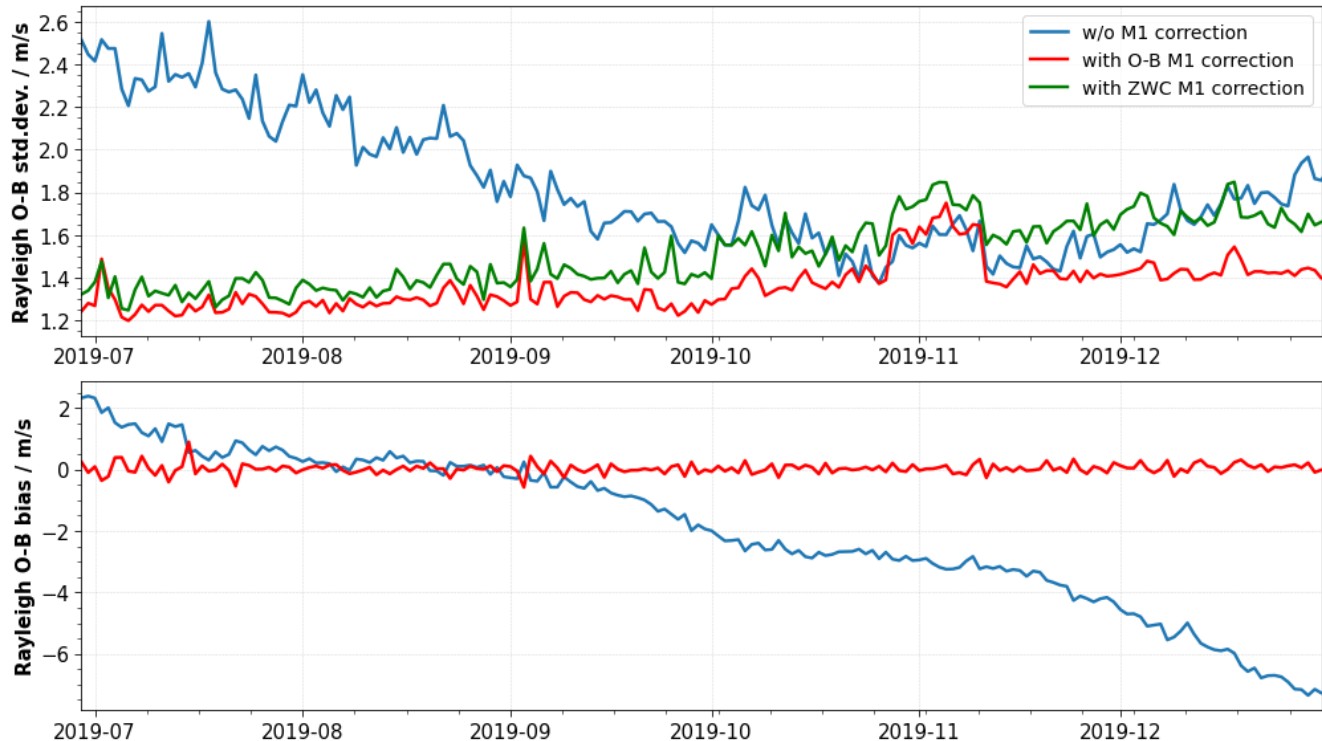

**Figure 13: Performance of the O-B (red) and ZWC M1 bias correction (green) methods for the period from 28 June to 31 December 2019. (Top) Time series of daily averages (15 orbits) of the standard deviation (std. dev.) of the Rayleigh clear E(O-B) HLOS values before (blue) and after the O-B (red) and ZWC (green) based M1 correction. (Bottom) Daily averages of the bias of the Rayleigh clear E(O-B) HLOS values before (blue) and after the O-B based M1 bias correction (red). Data from day N is used to predict the bias on day N+1.**




**Figure 14: Global distribution of the Rayleigh clear E(O-B) HLOS values (only ascending orbits) before (top) and after (bottom) the M1 bias correction. One week of data (111 orbits) from 15 to 22 August 2019 is shown. The gaps are due to calibration procedures such as "hot pixel" related calibration measurements or calibration measurements of the internal path.**

## 5 Summary

Already shortly after the successful launch of the Aeolus satellite in 2018, the operational assimilation of Aeolus wind products has been started at ECMWF in January 2020. A major milestone on the road to this achievement was the identification and correction of one of the most important systematic error sources for the Aeolus wind measurements. It was found that small





temperature variations of 0.3 °C across the primary M1 mirror of the Aeolus telescope lead to varying wind errors along the orbit of up to 8 m/s. This paper presents a detailed characterization of the telescope induced wind bias, describes the approach

to correct for this bias source and discusses the performance of the bias correction based on data between June 2019 and December 2019.

Our analyses have shown that the orbital variation of the Rayleigh wind bias changes over the seasons and, on top of that, strongly depends on the atmospheric scene. It turned out that the observed bias patterns are highly correlated with the temperatures measured at the primary telescope mirror. The telescope temperatures vary along the orbit as a result of changing

TOA short-and long-wave radiation of the Earth and the response of the telescope's thermal control system to that. The temperature changes affect the shape of the primary mirror which changes the focus of the telescope and it is assumed that this leads to a change of the angle of incidence of the incoming light at the spectrometers of the instrument and hence to a wind bias.

To correct for this effect a dedicated operational software was developed which describes the wind bias as a function of the

M1 telescope temperature in an MLR model. This approach is based on ECMWF model-equivalent HLOS winds as a bias-free reference and has been used successfully operationally at ECMWF since April 2020. The software uses 24 h of past data to derive the model fit coefficients and is updated twice per day. In this way, also the slowly drifting constant part of the wind bias can be corrected. It was demonstrated that the bias correction is capable of removing a large part of the M1 induced wind bias. In periods where the M1 influence on the wind bias is particularly strong, the bias correction can improve the

$STD(E(O-B))$-value of the Rayleigh clear HLOS winds by up to 53 % from 2.89 m/s to 1.36 m/s. The remaining residual bias variation is considered to be of mostly random nature and does not contain any obvious regular patterns. Moreover, the bias correction approach was also tested under special conditions during M1 optimization tests with changed thermal control law coefficients for the thermal control of the telescope. The results proved the reliability of the bias correction method even under these circumstances, paving the way for further in-orbit tests to improve the thermal control system of the telescope.

Despite the fact that on average the global bias of the u-components of ECMWF model w.r.t radiosonde observations is smaller than 0.3 m/s, the use of the NWP model as bias reference in the linear regression model is not ideal. Thus, this paper also presents the alternative of using ZWC winds as bias reference. The availability of ground returns is mainly restricted to polar regions with high surface albedo which makes the task of bias modelling based on ZWC more challenging. Hence, a downsized MLR approach with fewer independent variables is introduced. The results show that approach based on ZWC winds also

reduces most of the M1 induced bias variations and performs in most cases only slightly worse than the operational ECMWF model-based approach. However, it was also shown that the performance of the ZWC approach on average is 10.8 % worse than the ECMWF model-based bias correction, with maximum deviations of up to 25.6 %. Thus, it was decided to use ECMWF model winds as bias reference. Nevertheless, the goal is to remove the model dependence in the calculation of winds, so for the future it is planned to further improve the performance of the ZWC based approach and use it for upcoming reprocessing

campaigns or even in the NRT-processing of the Aeolus products. With the knowledge obtained during this study, it will be possible in principle to improve both the thermal design of the telescope and the optical setup to reduce the bias contributions



from the telescope temperature variation for a potential follow-on wind lidar mission. The goal would be to base the bias correction on measured ground-return speeds, as it was also initially foreseen also for Aeolus.

*Data availability.* The analysis in this paper is based on Aeolus Level 1B and 2B products. The Level 2B products are publicly available and can be accessed via the ESA Aeolus Online Dissemination System. The L1B products are processed in the framework of the second Aeolus reprocessing campaign and will become publicly available in 2021.

*Author contributions.* FW performed the data analysis and prepared the manuscript. MR, TK and LI largely contributed to the
development of the presented methods and the operational M1 bias correction. EC is the thermal engineer of Aeolus at ESA-ESTEC and provided valuable input to improve the understanding of the thermal control of the telescope. JdK is the developer of the L2B processor and adapted the software for the M1 bias correction. NO analysed the relationship between outgoing long-wave radiation and the M1 radial temperature gradient. OR is the scientific coordinator of the Aeolus DISC and supported the investigation. All authors reviewed the initial draft version of the manuscript and helped to continuously improve the
manuscript.

*Competing interests.* The authors declare that they have no conflict of interest.

*Disclaimer.* The presented work includes preliminary data (not yet publicly released) of the Aeolus mission that is part of the
European Space Agency (ESA) Earth Explorer Programme. This includes L1B wind products from the second reprocessing campaign which have not yet been publicly released. The used L1B products will become publicly available by summer 2021. The processor development, improvement and product reprocessing preparation are performed by the Aeolus DISC (Data, Innovation and Science Cluster), which involves DLR, DoRIT, ECMWF, KNMI, CNRS, S&T, ABB and Serco, in close cooperation with the Aeolus PDGS (Payload Data Ground Segment). The analysis has been performed in the frame of the
Aeolus Data Innovation and Science Cluster (Aeolus DISC).

*Special issue statement.* This article is part of the special issue "Aeolus data and their application (AMT/ACP/WCD inter-journal SI)".

*Acknowledgements.* The authors acknowledge the Aeolus Space and Ground Segment Operations teams, the Aeolus Data Innovation and Science Cluster as well as the technology centres at Airbus Defence and Space and ESA for their innovative spirit. The authors thank Gerhard Ehret for the internal review of this manuscript.




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
