# Peer review of "Correction of wind bias for the lidar on-board Aeolus using telescope temperatures"

_Atmospheric Measurement Techniques, 2021_

## Referee Comment (RC1)

**Review of manuscript "Correction of wind bias for the lidar on-board Aeolus using telescope temperatures" by Fabian Weiler et al**

**General comments**

The manuscript is well structured and addresses an important issue, i.e. systematic error, in the view of improving the impact of Aeolus HLOS winds in NWP. With this it is found as an important contribution to the Aeolus special edition and is as well in the scope of the AMT. The methods applied are well designed. Especially, apart from the very positive response of the bias correction methods presented, the potential weaknesses are discussed as well. Such is the usage of NWP winds, which inherently are biased. The addition of the ZWC based bias correction method is very appreciated in that regard. I don't have any major comments, although, below I list some minor questions and suggestions.

**Specific comments**

I have a question regarding the usage of O-B for computation of bias correction factors. I can not find in the manuscript if any quality control of O-B is provided before usage in the MLR method. Not all HLOS observations are valid, and also some might not be inconsistent with the model, which can affect the bias correction methodology. So my question: could you add what kind of QC is done before using O-B in MLR? For example it could be add in line 215.

In Line 225 you justify the usage of model as a reference for computation of bias correction factors. In particular you compare the model with radiosondes. This is of course meaningful only where radiosondes are present. It is more difficult to justify that over tropics, South hemisphere or even over oceans. These are exactly locations where we would like learn more from Aeolus. Maybe it would be valuable to mention this issue in the discussion section or conclusions?

Bias correction factors (AUX_TEL_12) are computed every 12h. How degraded is the bias estimate if estimated from data 12h in the past, i.e. how degraded is the bias correction estimate at the last 12h of this window? If you maybe estimated this? Line 382.

In the Summary section I suggest to provide explanations for abbreviations again (i.e. for MLR, ZWC, …). Since many readers will first read conclusions.

How fast is the reaction of heaters in M1 mirror, compared to the satellite ground speed? How this delay effects the bias correction factors?

How far (in days for example) in the future can bias correction still be used, if estimated by regression today?

**Technical corrections**

I noticed that in many cases articles "a", "an", "the" are missing, especially in combination with "as" (several are listed below).

Line 13/14: "at a global scale" could be replaced with "globally from space". In addition in Line 14, "into space" could be removed.

Line 17: It reads a bit confusing. Is "small" related to temperatures or fluctuations?

Line 21: The "short- wave radiation" is probably meant as a reflection of the sun shortwave radiation, not earth?

Line 22: "response" could be replaced with "related response", it reads better. In this regard the "to that" in Line 22 could be removed.

Line 23: "as bias reference" is a bit confusing, I would suggest replacing it with "as a reference to describe …"

Line 28: "However" is not appropriate. I would suggest replacing it with "Furthermore"

Line 29: "as" → "as a"

Line 30: "has" → "has a"

Line 47: I suggest to replace the order a bit: "The operational ... was started on 9 January 2020, followed by …"

Line 53: "This issue could …" should probably be: "This issue was successfully mitigated on ..."

Line 55: "unexpectedly large systematic error" maybe it would be valuable to explain that this is independent of the first source of error explained in the previous lines?

Line 85: Maybe it should be explained what is "terminator"? People from NWP (who are certainly interested in this study) may not be familiar with this term.

Line 99: I suggest splitting the sentence in two smaller sentences. "Thus ...determined. Afterwards, a projection …"

Line 99: "… plane the horizontal …" could be replaced with "… plane, the so-called horizontal ..."

Line 107: I suggest removing this sentence. It provides an additional complexity not needed anywhere in the manuscript.

Line 115: "and the wave front error" it reads as that telescope is made of wave front error? I don't understand.

Line 116: "the integration" what do you mean, what kind of integration?

Line 117: "specification" what does this specification mean? How does this affect the wind retrieval on Aeolus?

Line 125: "for each observations", what do you mean with observations?

Line 126: "TC-TC-23" → "TC-23"

Line 130: "the distance", but also the angle of the M2 in respect to M1 can be changed?

Caption: Figure 1. There should be a date/time of the last access present with the url.

Line 145: The Table 1 is not referenced anywhere in the text (I can not find it)? Also, since all of these are already described in the text, I don't see the need for adding a table.

Line 151: "fast cycle", what exactly is this cycle?

Line 159: maybe to replace "…" with "etc."?

Line 165: "as" → "as a"

Line 179: "9700 …", In line 174 it is stated that thresholds are set to 0. I don't understand how the 9700 LSB is associated with that thresholds? It should be better explained.

Line 196: "… of better quality", here I would suggest to add in addition, that "Mie-clear" winds are physically not really meaningful. These are mainly a result of the classification methodology.

Line 205: "Tco1279": maybe it would be valuable to add the meaning of this, i.e. cubic-octahedral spectral transform with spectral truncation of n=1279 (https://www.ecmwf.int/sites/default/files/elibrary/2016/17262-new-grid-ifs.pdf). Since many readers of this paper might not be from NWP.

Line 206: "0 h and 12 h": Isn't this up to 30 h, for situations when AUX_MET is missing (I think this information is present in L2B ATBD)?

Line 207: "interpolated", it is valuable to add the kind of interpolation.

Line 206: "The nearest neighbour": is this regarding the last sentence?

Line 209: Regarding the sentence "These differences … at a global scale". Here a reference on https://amt.copernicus.org/articles/14/2167/2021/ might be add, since there this is already thoroughly discussed.

Line 211: "… assimilation …" → "… data assimilation …"

Line 217: How exactly is this averaging done. Is this meant for Mie, since for Rayleigh the maximal accumulation is already 86.4 km? For Rayleigh accumulation length of observations can be of course smaller than 86.4 km, but I don't understand how can you average this back on the 86.4 km? Is this like some kind of interpolation along the track every 86.4 km?

Line 230: "as" → "as a"

Figurre 3: It would be more natural to invert x and y axes, so that altitude is on y axes. If possible? As well, is this before or after the quality control?

Figure 4: Is this globally? Meaning everywhere where radiosondes are present?

Line 241: "as" → "as a"

Line 248: "as" → "as a"

Line 253: I think there is one space too much in "360"

Line 272: "HISR" → "HIRS"

Line 365: "This is shown …." Is this consistent when repeated on other days?

Line 372: "… from to 2.89 m/s …" → "...from 2.89 m/s …"

Line 384: "As next step …" → "As a next step …"

Line 394: "To avoid  large …" Is this part of the scheme presented in Figure 10 or is done separatelly? Maybe to clarify this here.

Figure 9. In caption. "… dots indicates …" → "...dots indicate …"

Line 426: I suggest to remove "potential"

Line 429: "as" → "as a"

Line 436: "as" → "as a"

Line 440: The offset of 3 m/s. Is this consistent in time? How can this be removed, if it is not consistent in time?

Line 458: "confirm" → "to confirm"

Figure 11: caption. " … winds (green) and without …" → " … winds (green) without ….". As well "as" → "as a".

Line 468: What do you mean "… with different temperature set point conditions …"? What are set point conditions?

Line 476: What is "control law coefficients"? I don't understand.

Line 496: "… series …" → "… series, …"

Line 496: "The top plot shows the standard deviation …". Here I suggest to mention that this standard deviation is now statistics of daily averages.

Line 497: "… before and the after .." → "… before and after .."

Line 507: "… with more smaller …" → "...with smaller …"

Line 536: "as" → "as a"

Line 540: "reduced" → "reduces"?

Line 555: "… and ZWC (green) …" this can be removed, since it is explained in the first sentence of the caption.

Line 568: "of" → "of about"

Line 572: "… over the seasons …" → "… over seasons …"

Line 585: "STD(E(O-B))-value" probably there is some spaces missing here?

Line 587: Like above, it is not clear what the law coefficients stands for?

Lines 590-560: many cases of "as" → "as a"

Line 623: Maybe to change CNRS to  CNRS/Météo-France

---

## Author Comment (AC1)

**Response to Referee Comment #1 on**

*Correction of wind bias for the lidar on-board Aeolus using telescope temperatures*

The authors thank reviewer #1 for carefully reading the paper and providing very useful comments. In the following, referee comments are repeated in green and answers by the authors are provided directly below in black.

**General comments:**

The manuscript is well structured and addresses an important issue, i.e. systematic error, in the view of improving the impact of Aeolus HLOS winds in NWP. With this it is found as an important contribution to the Aeolus special edition and is as well in the scope of the AMT. The methods applied are well designed. Especially, apart from the very positive response of the bias correction methods presented, the potential weaknesses are discussed as well. Such is the usage of NWP winds, which inherently are biased. The addition of the ZWC based bias correction method is very appreciated in that regard. I don't have any major comments, although, below I list some minor questions and suggestions.

**Specific comments:**

I have a question regarding the usage of O-B for computation of bias correction factors. I cannot find in the manuscript if any quality control of O-B is provided before usage in the MLR method. Not all HLOS observations are valid, and also some might not be inconsistent with the model, which can affect the bias correction methodology. So, my question: could you add what kind of QC is done before using O-B in MLR? For example, it could be added in line 215.

For the calculation of the regression coefficients only valid L2B wind results, with product confidence flag set to true, are used. The Aeolus L2B products also contain HLOS error estimates for each wind result which are based on noise propagation from the signal-to-noise ratio of the useful signals. Maximum thresholds of 4 m/s and 8 m/s are used for the Mie and Rayleigh channel, respectively, following recommendations from ECMWF. This information was added to the text:

> 220 from June 2019 to December 2019 also includes this improvement.
> For the following analysis of the dependency of the wind bias on the M1 temperatures, a representative average O-B value "$E(O - B)$" is calculated from the L2B O-B values. For this only L2B wind results with the overall validity flag set to true are used. In addition, HLOS error estimates, reported for each L2B wind result, are used as quality criterion. Only Mie and Rayleigh wind results with HLOS error estimates smaller than 4 m/s and 8 m/s, respectively, are considered. As mentioned
> 225 above, the O-B values are available for each L2B wind result. However, to decrease the variance of the bias, the O-B values

In Line 225 you justify the usage of model as a reference for computation of bias correction factors. In particular you compare the model with radiosondes. This is of course meaningful only where radiosondes are present. It is more difficult to justify that over tropics, South hemisphere or even over oceans. These are exactly locations where we would like learn more from Aeolus. Maybe it would be valuable to mention this issue in the discussion section or conclusions?

You are correct. The comparison is restricted to regions where radiosondes and pilots are available. Figure 1 and 2 below show the coverage of radiosondes and pilots used for the comparison. It shows that the comparison is mainly restricted to land areas in the northern hemisphere, although there also

some measurements available in tropical regions (see Figure 2 around Indonesia).  This point was added to Sec. 2.4 of the manuscript:

235  It is obvious that the use of the NWP model for the bias correction introduces some NWP model bias dependency, since the ECMWF model wind biases are not zero. However, this is justified by the fact that on average the global bias of the model wind u-components w.r.t reference measurements based on radiosondes is relatively small. This is demonstrated in Figure 4 which shows the time series of daily averages of E(O-B) values of the ECMWF's u-component winds computed for radiosondes and pilots (radiosondes that only measure wind). It should be mentioned that the comparison is restricted to

240  locations where radiosondes and pilots are available, which is mainly above northern hemisphere land surface. Thus, it is difficult to accurately assess the model bias in the southern hemisphere or above oceans. -Nevertheless,  Figure 4  shows that dDuring all days the bias is clearly below 0.3 m/s, which is significantly smaller than the Aeolus M1 related bias (for the Rayleigh), justifying the choice of the ECMWF model as a reference for the bias correction.

[Figure]

*Figure 1: ECMWF data coverage of radiosondes for April 28, 2021.*

[Figure]

[Figure]

*Figure 2: ECMWF data coverage for pilots (radiosondes that only measure wind) and wind profilers for April 28, 2021.*

However, it should be mentioned that the bias of GNSS radio occultation profiles, which are distributed globally and are not bias corrected, is very low. These measurements act as anchor measurements for the whole data assimilation system. They may be measuring vertical profiles of mass information, however through 4D-Var this provides a strong constraint on the wind field. Also, there are globally distributed microwave radiances that are sensitive to humidity information, which again through 4D-Var leads to a strong constraint on the wind field. So, I would not expect wind biases to be that much worse away from radiosonde sites. The only exception is in the tropics e.g. in the UTLS near the ITCZ. There, biases between different operational NWP analyses can be quite large (see Figure 7, page 25 in https://www.ecmwf.int/en/elibrary/18014-advanced-monitoring-aeolus-winds).

Bias correction factors (AUX_TEL_12) are computed every 12h. How degraded is the bias estimate if estimated from data 12h in the past, i.e. how degraded is the bias correction estimate at the last 12h of this window? If you maybe estimated this? Line 382

The reason for the high update frequency is twofold. On the one hand, the instrument's sensitivity towards telescope temperature variations slowly changes with time. This manifests as a slow change of the model coefficients over time, showing the need for regular updates. On the other hand, it is necessary to capture the global bias drift induced by the drift of the internal reference of the instrument. This drift happens on shorter timescales than the first effect and has larger impact on the overall performance of the bias correction. It mainly affects the performance in reducing the average offset w.r.t to the ECMWF model.

The bottom plot of Figure 13 of the manuscript shows the performance of the correction for 24h updates. In this case, the average offset is always below 1 m/s HLOS. In operations, an update frequency of 12 h is used which further improves this result. The performance depends on the change of the internal reference response in a 12h interval. To quantify this effect for the analysed period, the change rate of the Rayleigh HLOS bias (blue curve in the bottom plot of Figure 13 in the manuscript) is calculated. First, a rolling mean of the Rayleigh HLOS bias with a window size of 24 hours is calculated,

corresponding to the 24h of data used for the regression. Next, binned averages using a duration of 12h are calculated and the difference of adjacent bins is analysed. On average, the difference, i.e. the remaining bias after the bias correction, is very small with 0.09 m/s. The maximum value (worst case) is 0.78 m/s which is still well within the systematic error requirements. This information was added to the manuscript:

> is changing over time. Moreover, this allows to capture the slowly drifting global average bias. Investigations have shown that the global average bias changes are due to a slow drift of illumination of the Rayleigh spectrometers in the internal path
>
> 410 (particularly for the FM-B laser). In order to capture this effect, it was decided to implement an intercept term $\beta_0$ (see Eq. 1) into the model which makes sure that mean of the model residuals, i.e. the $Mean[E(O - B)]$ value of the analyzed period, is zero. To avoid large constant bias offsets from the model, the mean winds are anchored to the ECMWF model twice per day and the AUX_TEL_12 generation is updated every 12 hours. The introduced bias offset depends on the change rate of the internal reference response in the 12h interval. For the analyzed period, the maximum change (worst case) was considerably
>
> 415 small with 0.78 m/s. To make sure that the sample size is large enough and the fit coefficients are derived with a sufficiently high enough accuracy, 24 hours of past data (~6500 samples) is used in the MLR.

Due to unexpected bias jumps related to signal jumps in the Rayleigh internal reference, the internal reference is not used anymore in the operational wind processing from 14th December 2020 onwards (time period not covered in the manuscript). Instead, the internal Rayleigh response has been fixed to a constant value which is possible as the M1 bias correction takes care of drifts linked to the atmospheric path of the instrument which would have been corrected by the internal reference. So, the high update frequency is also important to quickly correct for unexpected bias jumps associated with the atmospheric path of the instrument.

In the Summary section I suggest to provide explanations for abbreviations again (i.e. for MLR, ZWC, …). Since many readers will first read conclusions.

Agreed – the summary was changed.

How fast is the reaction of heaters in M1 mirror, compared to the satellite ground speed? How this delay effects the bias correction factors?

The thermistors are mounted on the rear side of the telescope. Thus, the reaction of the thermistors to environmental changes is limited by the conductive coupling through the M1 mirror thickness. In addition, the calculation of the heater duty cycle of each heating line for the thermal control loop is based on thermistor measurements is performed only every second. So, there is some delay. However, it is difficult to assess how this delay compares to the satellite ground speed. But looking at Figure 7 (top) of the manuscript, gives a hint about the reaction time of the thermistors. Based on the latitudinal displacement of the cold clouds of the ITCZ visible for ascending and descending orbits, it seems that there is a delay of a few 1000 km corresponding to a delay of a few minutes. More importantly, the bias correction is not affected by this delay as it is based on the actual M1 temperatures and not the corresponding TOA flux from the atmosphere.

How far (in days for example) in the future can bias correction still be used, if estimated by regression today?

The figure below compares the approaches of using fixed model coefficients with updating the model every 24 hours. The red curve indicates the performance using a fixed set of regression coefficients, obtained on day 1 of the analysed period, for the complete period. Due to the internal reference drift

the mean bias continuously increases (bottom plot) from close to 0 m/s at the beginning of the period to 10 m/s at the end of the period. The daily averages of the standard deviation of the Rayleigh bias of both approaches are comparable at the beginning of the period. However, approximately after month the performance of the model with fixed coefficients begins to degrade, showing the necessity of regular updates.

[Figure]

*Figure 3: Time series of daily averages (15 orbits) of the standard deviation (std. dev.) (top) and the bias (bottom) of the Rayleigh clear E(O-B) HLOS values after the O-B based M1 correction based. For the green curve data from day N is used to predict the bias on day N+1 The red curve is based on a fixed prediction model derived on day 1 (June 28).*

**Technical corrections:**

We would like to thank the reviewer for this thorough review and the large number of technical corrections, which significantly improved the manuscript.

Line 13/14: "at a global scale" could be replaced with "globally from space". In addition, in Line 14, "into space" could be removed

This was corrected in the manuscript.

Line 17: It reads a bit confusing. Is "small" related to temperatures or fluctuations?

The sentence was rephrased.

Line 21: The "short- wave radiation" is probably meant as a reflection of the sun shortwave radiation, not earth?

Yes, this is correct. The sentence was rephrased.

Line 22: "response" could be replaced with "related response", it reads better. In this regard the "to that" in Line 22 could be removed.

The sentence was changed accordingly.

Line 23: "as bias reference" is a bit confusing, I would suggest replacing it with "as a reference to describe …"

The sentence was changed accordingly.

Line 28: "However" is not appropriate. I would suggest replacing it with "Furthermore"

Yes, this is correct. The sentence was changed.

Line 29: "as" → "as a"

This was changed.

Line 30: "has" → "has a"

This was changed.

Line 47: I suggest to replace the order a bit: "The operational ... was started on 9 January 2020, followed by …"

The order of the sentence was changed accordingly.

Line 53: "This issue could …" should probably be: "This issue was successfully mitigated on ..."

This was changed.

Line 55: "unexpectedly large systematic error" maybe it would be valuable to explain that this is independent of the first source of error explained in the previous lines?

Yes, this is independent of the hot pixel issue. This was clarified in the text.

Line 85: Maybe it should be explained what is "terminator"? People from NWP (who are certainly interested in this study) may not be familiar with this term.

The terminator is the line dividing the Earth into sunlit and night-time areas. This information was added to the manuscript.

Line 99: I suggest splitting the sentence in two smaller sentences. "Thus ...determined. Afterwards, a projection …"

Agreed. This was changed.

Line 99: "… plane the horizontal …" could be replaced with "… plane, the so-called horizontal ..."

This was changed.

Line 107: I suggest removing this sentence. It provides an additional complexity not needed anywhere in the manuscript.

It was to decided to keep this information in the manuscript since the on-chip accumulation of charges to so-called "measurements" is a very important aspect in the processing chain.

Line 115: "and the wave front error" it reads as that telescope is made of wave front error? I don't understand.

It was split into two sentences.

Line 116: "the integration" what do you mean, what kind of integration?

The wave front error was determined after the final assembly of the instrument. This was clarified in the text.

Line 117: "specification" what does this specification mean? How does this affect the wind retrieval on Aeolus?

The WFE of the telescope determines the divergence of the transmitted laser beam in the atmosphere together with the laser divergence at the output of the laser. This divergence of the transmitted beam determines the footprint of the laser, and thus the signal budget together with the field-of-view of the telescope. Thus, it is a relevant contributor to the random error, in case that the transmitted laser beam is larger than the FOV.

Line 125: "for each observations", what do you mean with observations?

The observation granularity is introduced in Section 2.1 of the manuscript (line 110). 30 measurements are accumulated to one observation with a duration of 12s.

Line 126: "TC-TC-23" → "TC-23"

This was changed.

Line 130: "the distance", but also the angle of the M2 in respect to M1 can be changed?

No, it is not possible to change the angles between both mirrors by the implemented equal heating for the struts.

Caption: Figure 1. There should be a date/time of the last access present with the url.

Information about the last access was added to the figure caption.

Line 145: The Table 1 is not referenced anywhere in the text (I cannot find it)? Also, since all of these are already described in the text, I don't see the need for adding a table.

A reference to the table was added in the text.

Line 151: "fast cycle", what exactly is this cycle?

This is indeed misleading. The part of the sentence with "fast cycle" was removed.

Line 159: maybe to replace "…" with "etc."?

This was changed.

Line 165: "as" → "as a"

This was changed.

Line 179: "9700 …", In line 174 it is stated that thresholds are set to 0. I don't understand how the 9700 LSB is associated with that thresholds? It should be better explained.

In the operational processing of ground return winds, the thresholds for the minimum ground useful signal are set to 0. Thus, a large number of ground returns, possibly also low-quality data, is reported in the L1B products. But as the ground useful signal is reported as well in the product, it is possible to apply dedicated quality control to the ground return winds before doing further analysis. In this case, a minimum useful signal threshold of 9700 LSB is applied to the Rayleigh ground returns, ensuring the removal of gross outliers from the dataset. The paragraph in the text was rephrased accordingly.

Line 196: "… of better quality", here I would suggest to add in addition, that "Mie-clear" winds are physically not really meaningful. These are mainly a result of the classification methodology.

Yes, this is correct. This information was added to the text.

Line 205: "Tco1279": maybe it would be valuable to add the meaning of this, i.e. cubic-octahedral spectral transform with spectral truncation of n=1279 (https://www.ecmwf.int/sites/default/files/elibrary/2016/17262-new-grid-ifs.pdf). Since many readers of this paper might not be from NWP.

This information was added to the text.

Line 206: "0 h and 12 h": Isn't this up to 30 h, for situations when AUX_MET is missing (I think this information is present in L2B ATBD)?

The AUX_MET is produced every 12 hours and contains a forecast range up to 30 hours (0 to 30 hours). The above "usually in the forecast range 0 h and 12 h" is perhaps a bit confusing. It was trying to emphasise that the forecast data (which we call B) applied in the L2B product O-B statistics are usually short-range forecasts i.e. in the range 0-12 hours i.e. we are not using longer forecast ranges e.g. 20-30 hours, due to the twice per day update of the files - and an only 3-hour delay on delivery of the AUX_MET file in operational processing relative to the validity start of the file.  The shorter the forecast range the more accurate the forecast since the errors grow exponentially. The paragraph was rephrased.

Line 207: "interpolated", it is valuable to add the kind of interpolation.

It's a nearest neighbour interpolation. This information was added.

Line 206: "The nearest neighbour": is this regarding the last sentence?

Yes, this is the explanation of the interpolation.

Line 209: Regarding the sentence "These differences … at a global scale". Here a reference on https://amt.copernicus.org/articles/14/2167/2021/ might be add, since there this is already thoroughly discussed.

Thank you very much for pointing us to this reference, which was added here.

Line 211: "… assimilation …" → "… data assimilation …"

The sentence is correct. It's already "… data assimilation …".

Line 217: How exactly is this averaging done. Is this meant for Mie, since for Rayleigh the maximal accumulation is already 86.4 km? For Rayleigh accumulation length of observations can be of course smaller than 86.4 km, but I don't understand how can you average this back on the 86.4 km? Is this like some kind of interpolation along the track every 86.4 km?

For the Rayleigh channel the maximum accumulation length is 86.4 km which corresponds to the length of one L1B observation. However, also shorter accumulation lengths can occur. I agree that for cases with accumulation lengths close to 86.4 km, it's is more like a selection of Rayleigh wind results rather than an averaging. This is clarified in the text.

Line 230: "as" → "as a"

This was changed.

Figure 3: It would be more natural to invert x and y axes, so that altitude is on y axes. If possible? As well, is this before or after the quality control?

The axes were inverted. It's only showing wind results after the quality control described above in the specific comments section. This is information is added to the figure caption.

Figure 4: Is this globally? Meaning everywhere where radiosondes are present?

Yes, this is globally. The location of the radiosondes and pilots is in the plots above.
Line 241: "as" → "as a"

This was changed.

Line 248: "as" → "as a"

This was changed.

Line 253: I think there is one space too much in "360"

You are correct. This was changed.

Line 272: "HISR" → "HIRS"

Thank you. This was changed.

Line 365: "This is shown …." Is this consistent when repeated on other days?

Yes, the shape of the residual curve is comparable with other days.

Line 372: "… from to 2.89 m/s …" → "…from 2.89 m/s …"

This was changed.

Line 384: "As next step …" → "As a next step …"

This was changed.

Line 394: "To avoid large …" Is this part of the scheme presented in Figure 10 or is done separately? Maybe to clarify this here.

This is part of the scheme presented in Figure 10. Further clarification was added to the text.

Figure 9. In caption. "… dots indicates …" → "…dots indicate …"

This was changed.

Line 426: I suggest to remove "potential"

Agreed.

Line 429: "as" → "as a"

This was changed.

Line 436: "as" → "as a"

This was changed.

Line 440: The offset of 3 m/s. Is this consistent in time? How can this be removed, if it is not consistent in time?

This offset is quite consistent with time and is considered to be a result of the different calibration schemes used to process L2B wind results (AUX_RBC) and L1B ZWC winds (AUX_RRC). Information about the consistency with time was added to the text.

Line 458: "confirm" → "to confirm"

This was changed.

Figure 11: caption. "… winds (green) and without …" → "… winds (green) without …." As well "as" → "as a".

This was changed.

Line 468: What do you mean "… with different temperature set point conditions …"? What are set point conditions?

The thermal control of the telescope aims at keeping the telescope temperatures at a fixed temperature set point. This is explained in Section 2.2 of the manuscript. For the tests the set point was changed to test the influence on the radiometric performance of the instrument.

Line 476: What is "control law coefficients"? I don't understand.

The thermal control of the M1 mirror is based on a Proportional Integrational Differential (PID) control loop which controls the heating power applied to the TC sensors. To get the optimal response certain coefficients of the control algorithm can be varied. This information was added.

Line 496: "… series …" → "… series, …"

A semicolon was added.

Line 496: "The top plot shows the standard deviation …". Here I suggest to mention that this standard deviation is now statistics of daily averages.

This information was added to the text.

Line 497: "… before and the after ..." → "… before and after ..."

This was changed.

Line 507: "… with more smaller …" → "...with smaller …"

This was changed.

Line 536: "as" → "as a"

This was changed.

Line 540: "reduced" → "reduces"?

This was changed.

Line 555: "… and ZWC (green) …" this can be removed, since it is explained in the first sentence of the caption.

This was changed.

Line 568: "of" → "of about"

This was changed.

Line 572: "… over the seasons …" → "… over seasons …"

This was changed.

Line 585: "STD(E(O-B))-value" probably there is some spaces missing here?

No, this seems to be correct.

Line 587: Like above, it is not clear what the law coefficients stands for?

See comment above. An explanation was added to Sec. 4.1 of the manuscript.

Lines 590-560: many cases of "as" → "as a"

This was changed for multiple cases.

Line 623: Maybe to change CNRS to CNRS/Météo-France

Done.

---

## Author Comment (AC2)

**Response to Referee Comment #2 on**

*Correction of wind bias for the lidar on-board Aeolus using telescope temperatures*

The authors thank reviewer #2 for carefully reading the paper and providing valuable input. In the following, referee comments are repeated in green and answers by the authors are provided directly below in black.

**General comments:**

This is an excellent paper that describes empirical correction of Aeolus wind bias based on temperature gradients across the primary mirror. The work is important because correction of bias is an important consideration for assimilation of data into numerical forecast models. Two methods are investigated: one based on comparisons of measurements with the ECMWF model, and one derived from measurements of the velocity from ground hits of the transmitted laser pulse. The paper is well-organized and provides details on the correction methodology as well as performance of the correction methods described. Although the results are unique to Aeolus, and therefore are likely of somewhat limited impact for other instruments, the analysis showing the impacts of temperature gradients across the mirror and the conclusion that empirical corrections can be successfully applied are potentially important for addressing unanticipated problems in that crop up in future missions. Although I think the paper could be published as is, there are a few places in the text where a bit more detail and explanation might be useful to the reader. I leave the decision on whether to request these changes to the discretion of the editor.

**Specific comments:**

Line 207: It isn't clear to me how the 86 km averaging of the Rayleigh channel is taken into account when comparing the AUX_MET data with the Level 2B results. The text implies that the nearest neighbour from the model is compared to the L2B data, but the discussion seems unclear to me on issues such as 1) Are the O-B statistics comparing an 86 Km average with a single point from a 9 km grid-spaced data set, and 2) Is the level 2B HLOS measurement placed at the centre of the 86 Km swath? Perhaps I'm missing something here, but it seems that some clarification on the details of the comparison would be useful here.

The L2B processor uses a nearest neighbor approach in the horizontal dimension and just uses the closest profile provided in the AUX_MET file in the selected time window. In the vertical dimension a spline interpolation is used to get a value at the proper altitude. We do not do any area averaging and do not use something like an observation operator. More precisely, the centre-of-gravity (CoG) location of the L2B winds is used to derive the values from the AUX_MET for the O-B statistics. The horizontal CoG location of the L2B winds is determined by the CoG of the signals that were included in the accumulation. Due to the classification into clear/cloudy measurements this may deviate significantly from the center of the 86 km group length for the Rayleigh channel (or 14 km group length for the Mie channel). For the vertical location within the range bin the center position is considered.

ECMWF tested the impact of an averaging operator on the Rayleigh clear wind O-B and found it only improved the stdev(O-B) by 0.04 m/s relative to a point-like operator and didn't have any detectable bias improvement. This test was done using AUX_MET data. So, a point-like observation operator is considered to be sufficient for M1 T related bias correction. Note that the AUX_MET IFS profiles are provided along the orbit every 3 seconds (~21 km spaced); despite the underlying ECMWF model being run at higher resolution of Tc01279 (~9 km grid spacing). The grid-spacing of 9 km does not give a true

reflection of the model resolution however, the effective resolution has been estimated in the past to be ~4-8 times the grid spacing (https://www.ecmwf.int/en/elibrary/17358-effective-spectral-resolution-ecmwf-atmospheric-forecast-models). This effective resolution of 40-80 km probably explains the negligible improvement in O-B statistics by accounting for Aeolus' footprint (averaging). Further information about the O-B calculation was added to the manuscript:

> files is provided every 3 seconds along the orbit at 137 model levels interpolated (nearest-neighbor) to the Aeolus track. To
>
> 215 compute observation minus background (O-B) statistics, tThe L2B processor uses a nearest-neighbor approach in the horizontal and uses the closest profile in the AUX_MET file in the selected time window. In the vertical dimension a spline

> interpolation is used to get a value at the proper altitude. nearest neighbor of the AUX_MET data to the L2B wind results is used to compute observation minus background (O-B) statistics on a global scale. These O-B statistics differences have been used to analyze the systematic and random wind errors of the Aeolus observations at a global scale (Martin et al., 2021). The

Line 218: The authors should perhaps provide some evidence for the statement "O-B values are averaged over all range gates which is justified by the lack of altitude dependency". One can make a case that the physical effect that creates the temperature gradients won't change with altitude, but it isn't clear whether the statement is based on that assumption or that a comparison was used to make the case for the lack of altitude dependency.

The M1 temperature gradients only change from observation to observation. For a fixed observation the M1 temperatures are constant for all altitudes. Thus, there can be no altitude effect induced by changing M1 temperature gradients. This was clarified in the text as follows:

> can be performed. Afterwards, the O-B values are averaged over all range gates which is justified by the lack of altitude dependency of the M1 bias effect to yield the $E(O-B)$ value. This is justified by the lack of altitude dependency of the M1
>
> 230 bias effect as the measured M1 temperatures are constant over all altitudes. Figure 3 shows the typical distribution of the centre-of-gravity altitudes of the L2B Mie cloudy and Rayleigh clear wind results. This plot indicates that for the Mie channel large fraction of wind results in the lower altitudes contribute to the $E(O-B)$ statistics. In contrast, the Rayleigh wind results

Line 255: If the bias structure is strongly dependent on the atmospheric scene, that would appear to limit the effectiveness as the scene changes from day to day. I assume that the effects are a function of the time scale of the changes in cloudiness versus the temperature response of the mirror, but perhaps a bit more discussion here could be useful.

It is true that the observed bias pattern strongly depends on the atmospheric scene. However, the correlation of the bias with the M1 temperatures, and thus the underlying physical effect, does not change too much from day to day. It was found that the instrument's sensitivity towards telescope temperature variations only changes slowly with time. This manifests as a slow drift of the model coefficients with time. As a result, it is possible to train a model on day N and use it to predict the bias for day N+1.

The predictive capability, i.e. how far in the future the bias correction can still be used, is limited by the drift of the internal reference of the instrument and the slow changes of instrument's sensitivity (as mentioned above and discussed in Sec. 3.2 of the manuscript).

Line 268: This is the same issue as noted above. The time scale of the OLR changes would seem to be important if the results from the prior day are being used to correct the bias for a given measurement period.

See the response above.

Line 359: The meaning of the sentence "Note that for the reprocessing data from the same time period is used to derive the fit coefficient." isn't clear to me. Perhaps I missed an earlier reference to reprocessing.

For the reprocessing there is no need to predict the M1 induced bias ahead. The advantage for reprocessing is the availability of the complete data set for 24 h, while for NRT processing only the last 24 h are available. As a result, it is possible to apply the regression model to the same bunch of data that was used to train the model. This further improves the performance as no out-of-sample predictions with unseen data have to be performed. This was clarified in the text:

> 375  2019. Note that for the reprocessing data from the same time period is used to derive the fit coefficients. The big advantage for reprocessing is the availability of the complete data set which makes it possible to apply the MLR model to the same time period that was used to train the model. This even further improves the performance of the bias correction scheme as no predictions with unseen data have to be performed. The left scatter plot indicates the high correspondence between the model prediction and the measured bias values. This is demonstrated by the high $R^2$ value of 0.78. The right plot of Figure 8 is

Line 438: In the sentence "Note that the constant offset of about 3 m/s between O-B and ZWC values is due to the different calibration procedure between L1B and L2B winds and is not considered to be a problem for the bias correction since this offset could be corrected in the data processing", reference is made to the different calibration procedures. A reference of a bit of explanation would be useful here.

The calibration of L2B Rayleigh winds includes Rayleigh-Brillouin scattering correction, based on so-called AUX_RBC files (Dabas et al. 2008, Rennie et al. 2021 L2B ATBD)). The AUX_RBC file contains a look-up table for instrument Rayleigh responses as a function of atmospheric pressure and temperature. The AUX_RBC file is derived from an internal reference calibration measurement, representative for the internal path of the instrument. In contrast to that, the calibration of ground return winds is based on calibration measurements that are representative for the atmospheric path of the instrument in nadir mode (Reitebuch et al. 2018, L1B ATBD). So, differences in the frequency offset between the atmospheric and internal path are responsible for the observed offset between L1B and L2B winds. The references to the ATDBs and the paper by Dabas were added to the manuscript.

Line 443: Regression theory is not my specialty but some metric or reference for why 659 samples is not sufficient to use the original model might be useful here.

In case the sample size is small compared to number of covariates, overfitting can occur such that the regression model tends to describe the noise rather than the physical relationship in the data. In such a case, the capability of the model performing predictions with unseen data is drastically reduced. For

the ZWC approach, differently sized linear regression models were generated and for each model the predictive skills, i.e. the ability to perform out-of-sample predictions, were evaluated. Based on that, the presented regression model was found. This information was added to text:

> not considered to be a problem for the bias correction since this offset could be corrected in the data processing.
> In contrast to the MLR model defined in Eq.1 a slightly different approach is used to describe the ZWC winds as a function of the M1 temperatures. Due to the lower sample size a simplified model with fewer independent variables has to be used. In case the sample size is small compared to the number of model coefficients, overfitting can occur, meaning that the model tends to describe the noise rather than the physical relationship in the data. In such a case, the capability of the model performing predictions with unseen data is drastically reduced. To avoid this issue, different MLR model combinations were tested and for each combination the skill in predicting the bias was evaluated. -It was found that a grouping of the thermistors into two groups which describe the temperature at the outer and inner parts of the M1 mirror provides the best results: $G1 = mean(AHT27, TC20, TC21)$ and $G2 = mean(AHT24, AHT25, AHT26, TC18, TC19)$. The bias correction model is then described as follows:

Line 462: It seems the "and" before "without M1 correction" in the caption for Figure 11 could be eliminated.

Thank you. The caption was corrected for the revised version of the manuscript.

Line 477: Use of the dash to indicate the temperature range causes confusion when followed by a negative number in "0.3°C – -0.1°C". Perhaps another way to articulate the range could be employed.

This was changed in the manuscript. Now, words are used to describe the ranges: "0.3°C to -0.1°C".

Line 523: It isn't clear to me why it would not have been possible to observe the increase in random error without the bias correction. Perhaps a sentence of explanation here would be useful.

The daily averages of the M1-uncorrected STD(E(O-B)) values are dominated by the M1 effect. Figure 13 of the manuscript shows daily averages of the STD(E(O-B)) for Rayleigh clear HLOS winds before and after the M1 correction. The decrease of blue curve, showing the statistic before the M1 correction, could be misinterpreted as a decrease of the random measurement error. In fact, the decrease is related to the change of the M1 temperature conditions, having less impact on the bias at the end of the period compared to the beginning. Only after correcting for M1 effect, the "true" random error reveals. Following information was added to the text:

> of the accurate M1 bias correction, it would not have been possible to observe the increase of the random error based on wind error statistics as the M1 effect dominates the $STD(E(O − B))$ values. The daily averages of the uncorrected $STD(E(O − B))$

> values (blue curve in the top plot of Figure 13) show a decrease from the beginning of the period until October 2019, related to the changing seasonal influence of the M1 temperature induced bias. This decrease could be misinterpreted as a decrease of the random wind error. Only after correcting for the M1 effect, the true random error evolution reveals.

---

## Referee Report (RR1)

**Review of manuscript "Correction of wind bias for the lidar on-board Aeolus using telescope temperatures" by Fabian Weiler et al.**

**General comments**

Authors provided very good update on the manuscript taking into account previous comments and suggestions. I especially appreciate the very clear explanation of several minor questions. I don't have any additional comment or suggestion.

---

## Author Response (AR2)

**Response to Referee Comment #3 on**

*Correction of wind bias for the lidar on-board Aeolus using telescope temperatures*

The authors thank Mr. Hui Liu for carefully reading the paper and providing useful feedback. In the following, referee comments are repeated in green and answers by the authors are provided directly below in black.

**General comments:**

This paper describes a correction of the bias in Aeolus winds related to the M1 temperature. This makes assimilation of Aeolus winds with NWP model much more successful and leads to improved impact on NWP. NWP users of Aeolus winds would benefit from the details of the bias correction as described in the paper. As such, this paper deserves published.

**Specific comments:**

1. abstract, line 28: "the approach of using ECMWF model-equivalent winds is justified by the fact that the global bias of models u-component winds w.r.t to radiosondes is smaller than 0.3 m/s", This statement may not be representative here since the majority of globe is not covered by radiosondes. Actually, over the large part of remote oceans and lands, NWP models still have large (on the order of several m/s) uncertainty including biases, e.g., in the upper troposphere and lower stratosphere of the Tropics. This comment also applies to line 236-239.
However, the regression of O-B to M1 temperatures globally and from all vertical layers makes the M1 correction less sensitive to the considerable latitudinal and vertical layer varying biases or uncertainty between NWP models. It might be helpful to make this point clearer in the paper.

It is correct that the statement about the low model bias is difficult to justify in regions with low radiosonde and pilots density. This point is addressed in lines 237 to 244. It is confirmed, for the M1 bias correction altitude varying model bias should not be an issue, because all O-B values of a profile are averaged before the fitting. Moreover, global model averages obtained from 24 hours of observations are used which should mitigate the effect of localized model errors.

The following information was added to Section 2.4 of the manuscript:

240     locations where radiosondes and pilots are available, which is mainly above northern hemisphere land surface. Thus, it is difficult to accurately assess the model bias in the southern hemisphere or above oceans. Nevertheless, Figure 4 shows that during all days the bias is clearly below 0.3 m/s, which is significantly smaller than the Aeolus M1 related bias (for the Rayleigh), justifying the choice of the ECMWF model as a reference for the bias correction. To mitigate the influence of model wind bias, 24 hours of global model winds averaged over all altitudes are used in the M1 bias correction. On the one hand, this

245     makes sure that localized small-scaled model biases (e.g. in the tropics) appear only as noise source in the fit procedure. On the other hand, averaging over all altitudes ensures that any altitude varying model bias is not an issue.

2. line 241: Potential wind background uncertainty may be explored by comparing winds from major NWP models, e.g., ECMWF vs. NOAA/GFS. In remote regions, current NWP models still have large uncertainty.

Yes, it is correct that in remote regions, especially in the tropics, model winds can be largely biased. Figure 1 below, for example, indicates the difference of the wind vector between the ECMWF and Met Office mean analysis over a 7-day period in 2015 at 100 hPa. Such analysis indeed helps to identify problem regions. As mentioned above, we try to mitigate the influence of localized model errors by using 24 h of vertically averaged global model winds. It is the preferred solution to use model-independent ground return winds to avoid model dependency. But results showed that the performance of this approach is not yet stable enough for the operational processing and analysis will continue.

For the comparison of different models, the Aeolus CalVal includes the different Met centers that can use in their analysis their own meteorological input data to further quantify the impact of the differences. This is an ongoing effort in a bigger framework.

[Figure]

*Figure 1: Mean(ECMWF analysis) − mean(Met Office analysis) from 1st to 7th May 2015 for vector wind at 100 hPa. Two analyses per day: 00 and 12 UTC. Only wind vectors > 2 m/s wind speed are plotted to highlight the problem areas. Produced by ECMWF (Michael Rennie) - from https://www.ecmwf.int/en/elibrary/18014-advanced-monitoring-aeolus-winds.*

3. Figure 14 (bottom) is very interesting. I guess this is the O-B average over the entire vertical layer range (0-24km?). It will be helpful to provide this information in the figure caption. Also, it would be greater if the magnitudes and details of the remaining biases could be better visualized, e.g., some kind scatter plots with density distributions (vs. latitude and/or longitude).

Yes, the figure shows vertically averaged E(O-B) values at the L1B observation granularity. The determination of such is also explained in Section 2.4 of the manuscript. For the sake of clarity, further information was also added to the caption of Figure 14.
To better highlight the remaining bias, Figure 2 further below shows the residual bias as a function of the argument of latitude. The plot is based on the same data period as shown in Figure 14 of the manuscript. The plot reveals that the binned average (solid red line) is close to zero over the major part of the orbit. However, for the region with particular strong M1 temperature influence, i.e. at 230°

and 330° argument of latitude, remaining bias with a binned average of up to 1 m/s is visible. Despite the M1 bias correction being highly effective, the currently used regression approach still can be improved. However, testing more sophisticated regression models, such as random forests (Svetnik et al., 2003) or generalized additive models (Hastie and Tibshirani, 2014), is beyond the scope of this paper and could be considered for our future work. Thus, the following information was added to the summary of the manuscript:

return-based approach and use it for upcoming reprocessing campaigns or even in the near-real-time-processing of the Aeolus products. In addition, more sophisticated regression models, such as random forests (Svetnik et al., 2003) or generalized 635 additive models (Hastie and Tibshirani, 2014), will be tested to further improve the performance of the M1 bias correction. With the knowledge obtained during this study, it will be possible in principle to improve both the thermal design of the telescope and the optical setup to reduce the bias contributions from the telescope temperature variation for a potential follow-on wind lidar mission. The goal would be to base the bias correction on measured ground-return speeds, as it was also initially foreseen also for Aeolus.

[Figure]

*Figure 2: Rayleigh clear E(O-B) HLOS values (red points) after the M1 bias correction as a function of the argument of latitude. The solid red line indicates binned averages of the E(O-B) values using a bin size of 5° for the argument of latitude. One of week data from 15 to 22 August 2019 is shown.*

4. It might be helpful to explicitly mention in the abstract and conclusion that the M1 correction has little impact on Mie winds.

It was decided to add this information to the summary:

and long-wave radiation of the Earth and the response of the telescope's thermal control system to that. The temperature changes affect the shape of the primary mirror which changes the focus of the telescope and it is assumed that this leads to a change of the angle of incidence of the incoming light at the spectrometers of the instrument and hence to a wind bias. 605 Moreover, it was found that the sensitivity of the Mie bias on the M1 temperatures is ~10 times less than for the Rayleigh channel. To correct for this M1 temperature effect a dedicated operational software was developed which describes the wind bias as a

*References:*

Hastie, T. and Tibshirani, R.: Generalized Additive Models, in: Wiley StatsRef: Statistics Reference Online, American Cancer Society, https://doi.org/10.1002/9781118445112.stat03141, 2014.

Svetnik, V., Liaw, A., Tong, C., Culberson, J. C., Sheridan, R. P., and Feuston, B. P.: Random Forest:  A Classification and Regression Tool for Compound Classification and QSAR Modeling, J. Chem. Inf. Comput. Sci., 43, 1947–1958, https://doi.org/10.1021/ci034160g, 2003.